# Reorganization of brain structural networks in aging: A longitudinal study

Ana Coelho[1,2,3] (iD)   |   Henrique M. Fernandes[4,5]   |   Ricardo Magalhães[1,2,3]   |
Pedro S. Moreira[1,2,3]   |   Paulo Marques[1,2,3]   |   José M. Soares[1,2,3]   |
Liliana Amorim[1,2,3]   |   Carlos Portugal-Nunes[1,2,3]   |   Teresa Castanho[1,2,3]   |
Nadine Correia Santos[1,2,3]   |   Nuno Sousa[1,2,3]

[1]Life and Health Sciences Research Institute (ICVS), School of Medicine, University of Minho, Braga, Portugal

[2]ICVS/3B's, PT Government Associate Laboratory, Braga/Guimarães, Portugal

[3]Clinical Academic Center – Braga, Braga, Portugal

[4]Center for Music in the Brain (MIB), Aarhus University, Aarhus, Denmark

[5]Department of Psychiatry, University of Oxford, Oxford, UK

**Correspondence**
Nuno Sousa, Life and Health Sciences Research Institute (ICVS), School of Medicine, University of Minho, Campus Gualtar, 4710-057 Braga, Portugal.
Email: njcsousa@med.uminho.pt

**Funding information**
Fundação para a Ciência e a Tecnologia; European Regional Development Fund; FP7 Health; Fundação Calouste Gulbenkian; European Social Fund

## Abstract

Normal aging is characterized by structural and functional changes in the brain contributing to cognitive decline. Structural connectivity (SC) describes the anatomical backbone linking distinct functional subunits of the brain and disruption of this communication is thought to be one of the potential contributors for the age-related deterioration observed in cognition. Several studies already explored brain network's reorganization during aging, but most focused on average connectivity of the whole-brain or in specific networks, such as the resting-state networks. Here, we aimed to characterize longitudinal changes of white matter (WM) structural brain networks, through the identification of sub-networks with significantly altered connectivity along time. Then, we tested associations between longitudinal changes in network connectivity and cognition. We also assessed longitudinal changes in topological properties of the networks. For this, older adults were evaluated at two timepoints, with a mean interval time of 52.8 months ($SD = 7.24$). WM structural networks were derived from diffusion magnetic resonance imaging, and cognitive status from neuro-cognitive testing. Our results show age-related changes in brain SC, characterized by both decreases and increases in connectivity weight. Interestingly, decreases occur in intra-hemispheric connections formed mainly by association fibers, while increases occur mostly in inter-hemispheric connections and involve association, commissural, and projection fibers, supporting the last-in-first-out hypothesis. Regarding topology, two hubs were lost, alongside with a decrease in connector-hub inter-modular connectivity, reflecting reduced integration. Simultaneously, there was an increase in the number of provincial hubs, suggesting increased segregation. Overall, these results confirm that aging triggers a reorganization of the brain structural network.

**KEYWORDS**
aging, cognitive performance, diffusion MRI, network, white matter

Edited by Jeremy Hogeveen and Junie Warrington. Reviewed by Jenny Reick and Andrew Bender.

--------------------------------------------------------------

# 1 | INTRODUCTION

Human brain undergoes structural and functional changes during aging, even in the absence of disease (Damoiseaux, 2017; Grady, 2012; Hakun et al., 2015; Lockhart & DeCarli, 2014; Soares et al., 2014). According to the "disconnected brain" theory, originally proposed by Geschwind in 1965 (Geschwind, 1965), these alterations are thought to account for the cognitive decline that is observed during normal aging (Andrews-Hanna et al., 2007; Bartzokis, 2004; Fjell, Sneve, Grydeland, et al., 2016; Madden et al., 2017; O'Sullivan et al., 2001). Since cognitive functions rely on the communication between distinct functional subunits that are anatomically connected (Bressler & Menon, 2010; Bullmore & Sporns, 2009; Craddock et al., 2013; Park & Friston, 2013; van den Heuvel & Hulshoff Pol, 2010), the disruption of this communication (measured either as changes in structural or functional connectivity (FC) between different brain regions) in aging could be one of the potential sources of cognitive decline (Antonenko & Flöel, 2013; Salat, 2011; Zimmermann et al., 2016).

Network analysis has emerged as a tool to characterize brain's structural and functional organization (Bullmore & Sporns, 2009). In this context, brain is conceptualized as a complex network of inter-connected regions, and thus it can be modeled as a graph, with nodes defined as brain regions and edges as the structural/functional connections between regions. Graph theory analysis allows the extraction of quantifiable topological properties of networks that can clarify the organization and function of the brain network (Rubinov & Sporns, 2010). Functional networks can be constructed from resting-state functional magnetic resonance imaging, where FC reflects the temporal coherence in the blood-oxygen-level-dependent signal across brain regions (Damoiseaux, 2017). On the other hand, structural networks can be built from structural magnetic resonance imaging (MRI), where the structural connectivity (SC) measure is the structural covariance of gray matter volumes or cortical thickness (Bassett et al., 2008). Another option for determining SC is to use diffusion MRI which allows the estimation of white matter (WM) pathways, and in this case the SC measure will be the number of streamlines or the probability of connection between brain regions (Damoiseaux, 2017). Previous studies have revealed a number of non-trivial properties of the human functional and structural networks, such as small-worldness, modular architecture, hubs and cores, rich club structure, among others (Bullmore & Sporns, 2009; van den Heuvel & Sporns, 2011; Wang et al., 2010).

Normal aging induces a reorganization of brain's structural and functional networks, characterized as reduced global and local efficiency (Gong et al., 2009; Wu et al., 2012; Zhao et al., 2015), increased shortest path length and clustering coefficient (Otte et al., 2015; Sala-Llonch et al., 2014), reduced rich club organization (Zhao et al., 2015), modularity architecture reorganization (Betzel et al., 2014; Wu et al., 2012), and also a decrease in long-range connections accompanied by simultaneous increase in short-range connections (Andrews-Hanna et al., 2007; Cao et al., 2014; Sala-Llonch et al., 2014; Wu et al., 2012). Recent studies have emphasized the

### Significance

Normal aging is characterized by structural and functional alterations in the brain contributing to cognitive decline, with structural connectivity (SC) being the anatomical backbone for the communication between different functional subunits. Previous studies have suggested that the disruption of this communication contributes to the age-related deterioration observed in cognition, but most have focused on average connectivity of the whole-brain or in specific networks, such as the resting-state networks. Here, using a longitudinal design, we show that aging induces a reorganization of the brain structural network, that is characterized by connectivity decreases in intra-hemispheric connections and increases in inter-hemispheric connections, alongside with a reduction in integration and an increase in segregation. SC decreases were mainly due to loss of association fibers, an observation which is consistent with the last-in-first-out hypothesis.

role of brain network connectivity in cognitive performance in aging. Associations between network connectivity changes and multiple cognitive domains, such as, visuospatial reasoning, information processing speed, crystallized ability, executive function, and memory have been reported (Bernard et al., 2015; Fjell, Sneve, Grydeland, et al., 2016; Fjell, Sneve, Storsve, et al., 2016; Persson et al., 2014; Wen et al., 2011; Wiseman et al., 2018). Nevertheless, most of these studies focus on graph theory metrics that reflect the topological organization of brain networks or in the connectivity weight of the whole-brain or of specific networks, such as resting-state networks (RSNs). In this study, in addition to explore longitudinal changes of topological properties of brain structural networks during normal aging, we also inspect the existence of sub-networks that present significant age-related alterations in connectivity weight. To our knowledge, this is the first study to explore the presence of these sub-networks in aging. We hypothesized that aging will induce a reorganization of the brain structural network, characterized by disrupted connectivity in specific sub-networks, consistent with the findings of previous longitudinal diffusion MRI studies which demonstrated alterations in diffusion tensor imaging (DTI)-based measures as a function of age (Sexton et al., 2014; Vinke et al., 2018). Furthermore, we hypothesized that changes in brain SC would affect differently distinct WM tracts, consistent with the last-in-first-out hypothesis which posits that brain regions that reach full maturation later are more vulnerable to age-related atrophy (Raz, 1999) and this was already observed for WM tracts (Bender et al., 2016; Slater et al., 2019). Brain network reorganization will also be characterized by changes in topological features, as it has already been reported in some cross-sectional studies (Betzel et al., 2014; Gong et al., 2009; Otte et al., 2015; Zhao et al., 2015), but to date no longitudinal study using diffusion MRI has explored changes in brain structural

network topology. To test this, we construct structural connectomes for a group of older adults who were followed longitudinally, with a mean time interval of 52.8 months (SD = 7.24). We characterized longitudinal topological alterations in SC using advanced structural connectomics analysis. Furthermore, we tested associations between longitudinal changes in network connectivity and cognition.

## 2 | METHODS

### 2.1 | Ethics statements

The present study was conducted in accordance with the principles expressed in the Declaration of Helsinki and was approved by local and national ethics committees. The study goals and tests were explained to the participants and all gave informed written consent.

### 2.2 | Participants

The participants included in this study are part of a larger sample recruited for the SWITCHBOX Consortium project (www.switchbox-online.eu/), and are representative of the general Portuguese population with respect to age, gender, and education (Costa et al., 2013; Santos et al., 2013, 2014). Primary exclusion criteria were inability to understand the informed consent, participant choice to withdraw from the study, incapacity and/or inability to attend MRI sessions, dementia and/or diagnosed neuropsychiatric and/or neurodegenerative disorder (from medical records). Mini-Mental State Examination (MMSE) scores below the adjusted thresholds for cognitive impairment were also used as exclusion criteria. The adjusted thresholds were the following: MMSE score <17 if individual with ≤4 years of formal school education and/or ≥72 years of age, and MMSE score <23 otherwise (follows the MMSE validation study for the Portuguese population) (Guerreiro et al., 1994). These exclusion criteria were applied at both evaluations. Subjects were evaluated at two timepoints, with a mean interval time between first and last assessments of 52.8 months (SD = 7.24). At each evaluation, participants underwent an imaging session and a battery of neurocognitive/neuropsychological tests.

In the first assessment, 100 subjects were contacted for MRI screening. From these, one subject did not finish the diffusion acquisition and four subjects had brain lesions/pathology. In the last assessment, 55 subjects accepted to participate and underwent MRI acquisition protocol, but one did not finish the diffusion acquisition. A total of 51 individuals with data from both the first and last evaluations met all the inclusion criteria for this study.

### 2.3 | Neurocognitive assessment

A team of certified psychologists performed an identical battery of neurocognitive tests at both timepoints. This included the following tests previously validated for the Portuguese population: Stroop color and word test, selective reminding test (SRT) and MMSE. A previous report from our group examined the longitudinal measurement invariance of this set of cognitive tests (Moreira et al., 2018). Using confirmatory factor analysis, we observed that a two-factor solution encompassing (a) a general cognition and executive functioning dimension (EXEC) and (b) a memory dimension (MEM) was reliable over time. The MEM factor was comprised of long-term storage, consistent long-term retrieval, and delayed recall variables assessed with SRT. The EXEC factor was composed of the variables MMSE and Stroop parameters: words, colors, and words/colors. We obtained evidence of partial strong invariance which indicates an equivalence of the factorial structure and factor loadings for all the items, as well as of the intercepts of most items comprising this factorial solution. Thus, we estimated factor scores for each of these dimensions, based on the estimates for the model with strictest measurement invariance. The analytical pipeline was based on a maximum likelihood mean- and variance-adjusted estimator implemented with MPlus. The mean factor scores for the two dimensions were extracted for each participant.

### 2.4 | MRI data acquisition

All imaging sessions were performed at Hospital de Braga (Braga, Portugal) on a clinical approved Siemens Magnetom Avanto 1.5T MRI scanner (Siemens Medical Solutions, Erlangen, Germany) with a 12-channel receive-only head coil. The imaging protocol included several different acquisitions. For the present study, only the diffusion-weighted imaging (DWI) acquisition was considered. For this, a spin-echo echo-planar imaging sequence was acquired with the following parameters: TR = 8,800 ms, TE = 99 ms, FoV = 240 × 240 mm, acquisition matrix = 120 × 120, 61 2-mm axial slices with no gap, 30 non-collinear gradient direction with $b = 1,000$ s/mm$^2$, one $b = 0$ s/mm$^2$ and 1 repetition.

All acquisitions were visually inspected by a certified neuroradiologist, before any pre-processing step, in order to ensure that none of the individuals had brain lesions and/or critical head motion or artifacts that could affect the quality of data.

### 2.5 | MRI data pre-processing

Data were pre-processed using FMRIB Diffusion Toolbox (FDT) provided with the FMRIB Software Library (FSL v5.0; https://fsl.fmrib.ox.ac.uk/fsl/). First, DWI images were corrected for motion and eddy current distortions, followed by rotation of gradient vectors according to the affine transformations used to register each volume. Then, the first b0 volume of each subject was extracted and skull stripped, which generated a brain mask that was applied to the remaining volumes in order to remove non-brain structures. Finally, local modeling of diffusion parameters was performed using *bedpostx* algorithm which employs Markov Chain Monte Carlo sampling

to build up probability distributions of the diffusion parameters at each voxel, thereby allowing modeling of crossing fibers (Behrens et al., 2007). In addition, we also extracted the levels of head motion in a diffusion scan, using FSL tools, for all subjects at both timepoints. We then sought to determine if these values were associated with age (Figure S1) or if they were different between timepoints (Figure S2). Since there was no significant correlation with age neither differences in head motion levels between assessments, we thus concluded that there was no need to account for this variable in subsequent statistical analyses.

## 2.6 | Network construction

Network nodes were defined as the 90 regions of the Automated Anatomical Labeling (AAL) template. These regions were normalized to each subject native diffusion space. This was done by applying the inverted affine transformation from diffusion space to Montreal Neurological Institute (MNI) space. Probabilistic tractography was used to estimate connections between nodes (i.e., edges). This was accomplished using *probtrackx2* algorithm from FDT toolbox. 5,000 streamlines were sampled from each voxel in the seed mask. This resulted in a SC matrix, for each subject, representing the number of streamlines leaving each seed mask and reaching any of the other regions. This matrix was normalized by first dividing each line by the waytotal value (i.e., the total number of generated tracts not rejected by inclusion/exclusion mask criteria) and then dividing by the maximum SC value of each individual, in order to have connectivity values between [0, 1]. Each element of this matrix, $P_{ij}$, represents the connectivity probability between region $i$ and region $j$. Since tractography is dependent on seeding location, the connectivity probability from i to $j$ is not necessarily equal to that from $j$ to i. Still, these two probabilities are highly correlated across the brain for all participants. Thus, we defined the undirected connectivity probability as the average of these two probabilities, $P_{ij}$ and $P_{ji}$, which originated an undirected connectivity matrix. At the end of this process, a 90 × 90 symmetric connectivity matrix for each subject was obtained. A threshold set to 1% of the strongest connection was then applied to each subject's SC matrix, in order to remove spurious connections. An additional threshold was applied, based on a consistency-based thresholding technique. This method measures the consistency of edge weights across subjects and retains the most consistent ones, with the goal of reducing the false positives in group-averaged connectivity matrices (for a description of the method see (Roberts et al., 2017)). It was proven to preserve more long-distance connections, than the traditional weight-based thresholding, which often removes such connections since, in general, they represent weak edges. In this work, we applied consistency-based thresholding at 30% density (the same density used in (Roberts et al., 2017)) and then we performed a validation of the method by analyzing the connections that were removed. Specifically, we analyzed the connections that were removed in each individual after applying the threshold, both

the number of connections removed (Figure S3) and the strength of these connections (Figure S4). Furthermore, the consistency-based threshold method generates a group consistency mask that is then applied to each subject's SC matrix, in order to retain only the most consistent connections. Thus, we also analyzed the connections that were present in this group consistency mask but were not present in all subjects SC matrices. Once again, we evaluated the number of connections that were not present in all subjects (Figure S5) and their strength (Figure S6). We can observe that, in each subject, a small percentage of connections is removed (Figure S3) and they represent mostly weak connections (Figure S4). Also, for each subject, the connections from group consistency mask that are not present in its SC matrix do not represent more than 50% of all group consistency links (Figure S5) and once again, the majority of these edges are characterized by low connection strength (Figure S6). Together, these results support the use of this threshold technique to remove spurious connections.

Given that our sample covers a 30-year age range and the rate of WM change is known not to be homogeneous across age (Sexton et al., 2014; Westlye et al., 2010), we performed an additional analysis to evaluate the potential impacts of both age and sex on our estimations of SC. To perform this, we analyzed the levels of intra- and inter-timepoint consistency (TC) in the resulting signatures of individual SC, that is, how consistent are the patterns of estimated SC across all subjects in a timepoint, as well as between timepoints. To do this, we used the following two strategies for evaluating TC in SC.

### 2.6.1 | TC-I

Intra-TC measured as the Pearson's correlation between each subject's SC and timepoint mean SC (considering upper diagonal matrix elements). The resulting r values were *z*-transformed (Fisher-Z transformation) before averaging and converting (inverse of Fisher-Z) the resultant TC back to r scale. This value represents the within-TC, that is, for each timepoint, how well all subjects' SC correlate with the timepoint's average SC.

### 2.6.2 | TC-II

Intra-TC measured as the distribution of Pearson's correlations between all possible pairs of subjects in a timepoint. The resulting distribution of all pairwise (pairs of subjects) SC comparisons is represented as a histogram. This indicates how well SCs in a timepoint correlate with each other. Inter-TC was also assessed by considering all subjects as part of the same timepoint.

## 2.7 | Graph theoretical analysis

Brain networks can be described in terms of its topological organization, using graph theory measures. Brain Connectivity Toolbox

(https://sites.google.com/site/bctnet/) was used to extract these metrics. The following local and global measures were computed.

## 2.7.1 | Degree

The degree of a node $k_i$, in a binary undirected network, is the number of links connecting node $i$ with the other $j = 1 ... N - 1$ nodes:

$$k_i = \sum_{j \neq 1} A_{ij}$$

where $A$ is the adjacency matrix.

The mean degree of an undirected network is the average of all node degrees:

$$k = \frac{1}{N} \sum_{i=1}^{N} k_i$$

## 2.7.2 | Connection density

The connection density of a network is the proportion of the actual number of edges in the network relative to the total possible number of connections. For an undirected network with $N$ nodes without self-connections, the total number of possible connections is given by $N(N - 1)/2$. Thus, the connection density, $\kappa$, of an undirected network can be measured as:

$$\kappa = \frac{2E}{N(N - 1)}$$

where $E$ is the total number of edges in the adjacency matrix.

## 2.7.3 | Global efficiency

Global efficiency is a measure of integration that reveals how efficiently information can be exchanged between nodes. It is defined by the mean of the inverse shortest path length, $l_{ij}$, between each pair of nodes:

$$E_{glob} = \frac{1}{N(N - 1)} \sum_{i \neq j} \frac{1}{l_{ij}}$$

## 2.7.4 | Nodal efficiency

Nodal efficiency measures how well a node is integrated within the network via its shortest paths, that is., how well a given node connects to all other nodes in the network. It is defined as the mean of the inverse shortest path length, $l_{ij}$, between a given node and all other nodes in the network:

$$E_{nodal}(j) = \frac{1}{(N - 1)} \sum_{i} \frac{1}{l_{ij}}$$

## 2.7.5 | Local efficiency

Local efficiency reflects globally how information is exchanged within the neighborhood of a given node. It is defined as the average nodal efficiency:

$$E_{local} = \frac{1}{N} \sum_{i} E_{nodal}(i).$$

## 2.7.6 | Characteristic path length

The characteristic path length, $L$, is the mean shortest path length between all possible pairs of nodes in a network. The shortest path between nodes $i$ and $j$ is equal to the minimum number of connections or the minimum cost needed to connect nodes $i$ and $j$, where connection cost is defined as the inverse of connection weight. So, the characteristic path length is defined as:

$$L = \frac{1}{N} \sum_{i} l_i = \frac{1}{N(N - 1)} \sum_{i \neq j} l_{ij}$$

where $l_i$ is the average shortest path length from node $i$ to all other nodes in the network and $l_{ij}$ is the shortest path length from node $j$ to node $i$.

## 2.7.7 | Clustering coefficient

The clustering coefficient is measured as the fraction of closed triangles that are connected to node $i$, relative to the total number of possible closed triangles between $i$'s neighbors. It is a measure of local interconnectivity in a network and is calculated as follows:

$$Cl = \frac{1}{N} \sum_{i \in N} \frac{2t_i}{k_i(k_i - 1)}$$

where $k_i$ is the degree of node $i$ and $t_i$ is the number of closed triangles attached to $i$.

## 2.7.8 | Small-world index

Networks with high clustering and low average shortest path length are considered small-world networks. This is quantified by the index $\sigma$, that is a ratio of the normalized clustering coefficient and shortest path length. The normalization of these measures is done by dividing their empirical value by the average measure of an ensemble of randomized networks that preserves the degree distribution of the original network. When $\sigma > 1$, the network is considered to present small-world properties.

$$\sigma = \frac{Cl/Cl_{rand}}{L/L_{rand}}$$

where $CI_{rand}$ is the average clustering coefficient of the randomized networks and $L_{rand}$ is the average shortest path length of the randomized networks. In this work, we generated an ensemble of 100 randomized networks.

In addition, the following topological features were also assessed.

## 2.7.9 | Modularity

Modularity quantifies the degree to which nodes of a network may aggregate into densely connected non-overlapping modules or communities (Fornito et al., 2016). Nodes within a community are more strongly connected with each other than with nodes outside this community. Thus, the optimal community structure will be the partition of the network that maximizes intra-module connectivity and minimizes inter-module connectivity. The index of modularity, Q, is given by the difference between the empirical degree of intra-module connectivity and the degree expected by chance (Fornito et al., 2016). The optimal community structure can be found by searching for the partition that maximizes Q. One popular algorithm used to find the optimal partition is the Louvain algorithm and, shortly, this is how it works: first, it starts with all nodes in a distinct module, then it chooses a node at random and merges it with the module that produces the largest gain in Q, these steps are repeated until no additional gains in Q are possible (Blondel et al., 2008). Given that at each iteration, nodes are chosen randomly, running the algorithm multiple times can lead to different solutions. Also, another limitation is the so-called degeneracy problem, that can cause the existence of large number of different solutions, since there is not a clear global maximum of Q (Good et al., 2010). To circumvent this problem, we ran the Louvain algorithm 10,000 times and selected the partition having the higher number of occurrences in the set of 10,000 partitions, that is, the partition that was more consistent. To compare the optimal community structures found at each timepoint, we defined a similarity metric. For each module in a partition, we found the module in the other partition that was more similar to this one (by finding the maximum of the number of shared regions divided by the total number of regions in the two modules). The similarity metric was then calculated as the mean of the maximum values for each module. Values close to 1 indicate higher similarity between the partitions.

Additionally, we characterized the overlap between each module of the optimal partition and RSNs. For this, we calculated a matrix where each entry represents the percentage of intersection between all anatomical regions in a module and a given RSN, normalized by the total intersected volume between all regions of the anatomical atlas and each of the RSNs. The anatomical atlas used was the AAL as it was also used to construct the SC matrices, and the RSN atlas used was the parcellation into seven RSNs from (Yeo et al., 2011).

## 2.7.10 | Hubs

Hubs can be defined as nodes with high regional efficiency ($E_{nodal}$) (Achard & Bullmore, 2007). Specifically, for each node, if the normalized $E_{nodal}$ (divided by the mean $E_{nodal}$ of all nodes) is larger than the normalized mean $E_{nodal}$ of all nodes of the network plus one standard deviation (SD), the node is considered a hub (Lo et al., 2010).

Furthermore, we analyzed the topological roles of nodes in the communication within and between modules. This allowed the classification of nodes into provincial and connector hubs. The definition of these roles is described below.

*Provincial Hubs* are nodes with high within-module degree z-score (greater than the mean plus SD of all nodes) and low participation coefficient (PC ≤ 0.3). Positive values of within-module degree z-score indicate high (above the average) intra-module connectivity, and thus higher values of this measure suggest that the node plays a central role in intra-modular communication. PC compares the number of connections of a node with other nodes in different modules, to the total number of connections to other nodes in the same module. Values close to one indicate that the edges of a node are distributed uniformly across modules while a value of zero means that all edges of the nodes are limited to its own module. Thus, provincial hubs are characterized by comprising most of their connections within their own module (Fornito et al., 2016).

*Connector Hubs* were also defined as nodes with high within-module degree z-score and high (PC > 0.3). This means they have many connections with other modules, and thus play a key role in inter-modular communication (Fornito et al., 2016).

## 2.8 | Statistical analysis

Statistical comparison of the SC matrices between first and last assessments, at the edge level, was performed by applying a paired sample t-test with SC as the dependent variable and time of evaluation as independent variable. The obtained SC networks are comprised of 90 nodes, yielding a total number of possible edges of 4,005 (90*89/2). Testing the hypothesis of interest at the edge level, therefore poses a multiple comparisons problem. In order to increase the statistical power of the analysis, we used the network-based statistics (NBS) procedure implemented in the NBS toolbox (https://sites.google.com/site/bctnet/comparison/nbs). This is a non-parametric statistical method that allows the identification of significantly different sub-networks, while controlling for the family-wise error rate (FWER) (Zalesky et al., 2010). First, it independently tests the hypotheses at every connection in the network and threshold the ones exceeding a user defined primary threshold, then it identifies sub-networks constituted by interconnected edges that survived the primary threshold. The significance of these sub-networks is then calculated by comparing their sizes to the distribution of the size of sub-networks obtained through random permutations of the original hypothesis. It is important to note that the primary threshold only affects the sensitivity of the method and thus, FWER is assured independently of this threshold. In this study, the primary threshold was set to $F = 17.0$, which was the maximum threshold that detected a unique significant connected component having more than two

connections (Figure S8). Longitudinal changes in SC detected with NBS are represented by significantly connected components at a corrected level of $p < 0.05$ FWER corrected.

Additionally, we extracted, for each subject, the mean connectivity values of the significant component resulting from the NBS approach. We analyzed these values separately for the connections with increases in connectivity between timepoints, the connections with decreases and both types of connections. We then examined, for these three types, the values of mean connectivity of all connections, intra-left, intra-right, and inter-hemispheric connections. Moreover, we tested potential associations between the connectivity values of these networks and cognitive scores of MEM and EXEC. The rmcorr R package (https://cran. r-project.org/web/packages/rmcorr/) was used to compute a repeated measures correlation coefficient between each sub-network and cognitive score. This coefficient, unlike simple correlation, does not violate independence assumptions nor requires averaging the data and thus is suitable to use with repeated measures data (Bakdash & Marusich, 2017). The $p$ values of all correlations were corrected for multiple comparisons, using the false discovery rate (FDR) method.

The comparison of graph measures between timepoints was performed using paired sample $t$-tests and $p$ values were corrected for multiple comparisons, using the FDR method. In addition, we analyzed, for each timepoint, the network fingerprints of inter-modular (global and connector-hub driven) and intra-modular connectivity. The same method of analysis as described in (Fernandes et al., 2019), was applied in this work. In summary, modular connectivity strength was defined as the degree (total number of connections) of all nodes constituting a module. To quantify this connectivity at both timepoints, a reference scheme of community structure was chosen based on the mean score of community-structure goodness-of-fit. Then, matrices of inter-modular and intra-modular connectivity were created for both timepoints.

## 2.9 | WM tracts analysis

After identifying sub-networks with significant longitudinal changes, using the NBS approach described before, we performed an additional analysis designed to identify the WM tracts that are responsible for connecting the brain regions comprising the identified sub-networks. For this, we used streamline density maps obtained with probabilistic tractography. These maps represent the number of streamlines reaching each voxel and one map is generated for each seed region. So, we first normalized the streamline density maps of each subject to the MNI space using the affine transformation computed previously, and then applied a threshold of 1% of the maximum number of streamlines to remove spurious connections (same threshold as applied to construct the SC matrices). Next, we extracted WM tract masks from the JHU WM tractography atlas (Hua et al., 2008; Wakana et al., 2007) and computed the mean intensity (i.e., mean number of streamlines) of the overlapping region between each of these WM tracts and the streamline density map

**TABLE 1** Basic demographic characterization of the study's cohort

|  | Mean ± *SD* (range) |
| --- | --- |
| *N* (females/males) | 51 (26/25) |
| Age at baseline (years) | 63.5 ± 7.41 (51–82) |
| Age at follow-up (years) | 68.0 ± 7.25 (55–86) |
| Interval (months) | 52.8 ± 7.24 (45–73) |
| Education (years) | 5.98 ± 3.97 (0–17) |
| F-MEM at baseline | 0.24 ± 0.98 (−1.51–2.23) |
| F-EXEC at baseline | 0.20 ± 1.01 (−2.46–1.72) |
| F-MEM at follow-up | 0.063 ± 1.00 (−1.64–2.67) |
| F-EXEC at follow-up | 0.098 ± 0.99 (−1.90–2.05) |

Abbreviations: F- EXEC, mean factor scores for the general cognition and executive function composite dimension; F-MEM, mean factor scores for the memory composite dimension.

of each region. We repeated the process for each subject and each timepoint, then we grouped all this information in two matrices (one for each timepoint) that represent the mean intensity values of the overlap between each seed region and each WM tract averaged across subjects. Then, we applied a threshold of 5% of the maximum value to each matrix. Finally, we calculated the proportion of change between timepoints by computing the difference between the matrices of the last and first timepoints and then dividing by the matrix of the first timepoint. To identify WM tracts connecting a pair of regions, we inspected the proportion of change matrix and we selected WM tracts that had, for both brain regions, negative or positive values if that connection represented an SC decrease or an SC increase, respectively. In case regions shared multiple WM tracts, we chose the tract with the highest mean intensity value at both timepoints. When there was not any common tract, we also chose based on the mean intensity values at both timepoints instead of the proportion of change.

## 3 | RESULTS

### 3.1 | Sample characterization

Table 1 shows the demographic characterization of the participants included in this study. In summary, mean age at baseline was 63.5 years (range, 51–82 years) and mean interval between evaluations was 52.8 months (range, 45–73 months). Interval time was not significantly associated with age at baseline ($r = −0.12$, $p = 0.41$). The sample was balanced for sex (51% females, 49% males) and they did not differ with respect to interval time ($t(30) = 0.14$, $p = 0.89$). Mean education level was 5.98 years (range, 0–17 years). Regarding memory, at baseline, the mean factor score was 0.24 (range, −1.51–2.23) and at follow-up it was lower, with a mean value of 0.063 (range, −1.64–2.67). EXEC scores also decreased between assessments, with a mean value of 0.20 (range, −2.46–1.72) at baseline and 0.098 (range, −1.90–2.05) at follow-up.

## 3.2 | Timepoint consistency

High levels of intra- and inter-TC were found for both timepoints in the estimation of whole-brain SC (Figure S7). We thus concluded that potential bias due to age and/or sex did not have a significant impact on the estimation of SC so that the inclusion of additional confounds for these variables in the statistical analysis was not necessary.

## 3.3 | SC longitudinal changes

From first to last timepoint there were significant changes in SC in a brain sub-network ($p < 0.001$), comprising 16 connections, where 9 correspond to decreases and 7 to increases in SC between timepoints (Figure 1). Analyzing the individual connections of this network, we observe that the connections with longitudinal decreases in connectivity are composed by three intra-left, five intra-right, and one inter-hemispheric connections. On the other hand, the connections with increasing connectivity are constituted by two intra-left and five inter-hemispheric connections. The summary of the connections is present in Table 2.

Next, we examined the mean connectivity values of the significant sub-network (Figure 2). In the network with all connections, we observe an overall decrease in connectivity. When examining the three types of connections (intra-left, intra-right, and inter-hemispheric) we see that connections within the same hemisphere exhibit a decrease between the two timepoints, with a more pronounced decrease for the right hemisphere, while inter-hemispheric connections show an increase along time. In the network of increases, inter-hemispheric connections are the major contributors for this increase, while intra-hemispheric connections have lower connectivity values. In the network of decreases, most of the decrease in connectivity is due to connections within the right hemisphere, while intra-left and inter-hemispheric connections present lower connectivity values. Rates of change for the different sub-networks are reported in Table 3.

Regarding associations between mean connectivity values and cognition, only the correlation between the network with increases and EXEC was significant, although it did not survive the multiple comparisons correction (Table S1). In order to verify if the multiple comparisons correction was too conservative, we calculated the bootstrap confidence interval of the correlation coefficient using 10,000 draws. We obtained the following result: $r = 0.31$, 95% CI [−0.40, −0.18]. Thus, we estimate with 95% confidence that the true correlation coefficient between the network with increases and EXEC is between −0.40 and −0.18. Analyzing this association, we see that higher values of SC in this network are related to lower values of EXEC (Figure S9).

## 3.4 | WM tracts analysis

Figures 3 and 4 present the proportion of change between timepoints in the mean number of streamlines encompassed by the volume of overlap between each WM tract and seed region of the sub-network with decreases (Figure 3) and the sub-network with increases (Figure 4). The values of the mean number of streamlines for each timepoint are displayed in Figures S10 and S11. For the regions of the sub-network with decreases, connections were composed of association (anterior thalamic radiation, uncinate fasciculus, superior longitudinal fasciculus, cingulate gyrus part of cingulum bundle) and commissural fibers (forceps minor). While for the sub-network of increases, connections were attributed to all types of fibers, namely, association (inferior fronto-occipital fasciculus, cingulate gyrus part of cingulum bundle), commissural (forceps minor), and projection fibers (corticospinal tract). There was one special case in the network of increases that were the connections involving the left middle cingulate cortex. This region had only decreases in the mean number of streamlines, so we chose the WM tract with the highest value in both timepoints that was the left cingulate gyrus part of the cingulum. Information of which WM tract connects each pair of regions along with mean number of streamlines values is summarized in Table 4.

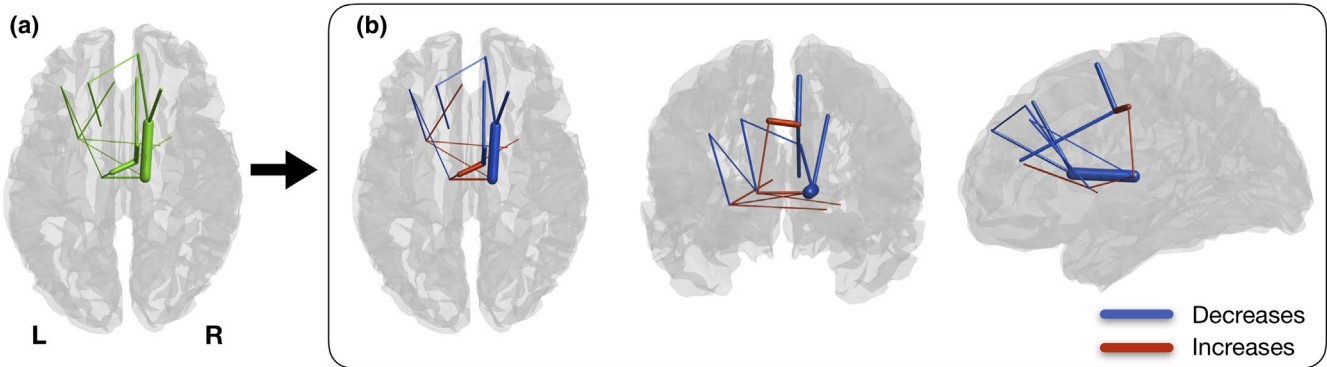

**FIGURE 1** Significant changes in structural connectivity between timepoints. (a) Binarized version of the connected component of significantly altered structural connectivity. (b) Weighted version of (a), with edge thickness representing the amplitude of differences. Blue represents decreases in connectivity strength between timepoints and red represents increases. Connections with decreases are mostly intra-hemispheric, while most of the increases are composed of inter-hemispheric connections. Both increases and decreases are mainly composed by links between subcortical and frontal regions [Color figure can be viewed at wileyonlinelibrary.com]

**TABLE 2** Description of the connections comprising the connected component of significant structural connectivity differences between timepoints ($p < 0.001$)

| Area 1 | | Area 2 | | | | | |
|---|---|---|---|---|---|---|---|
| N | Name | N | Name | Difference | Intra-left | Intra-right | Inter-hemispheric |
| *Increases* | | | | | | | |
| 77 | Thalamus L | 74 | Putamen R | 0.008 | 0 | 0 | 1 |
| 78 | Thalamus R | 73 | Putamen L | 0.009 | 0 | 0 | 1 |
| 76 | Pallidum R | 73 | Putamen L | 0.009 | 0 | 0 | 1 |
| 77 | Thalamus L | 33 | Cingulum Mid L | 0.013 | 1 | 0 | 0 |
| 73 | Putamen L | 31 | Cingulum Ant L | 0.013 | 1 | 0 | 0 |
| 78 | Thalamus R | 77 | Thalamus L | 0.019 | 0 | 0 | 1 |
| 34 | Cingulum Mid R | 33 | Cingulum Mid L | 0.044 | 0 | 0 | 1 |
| *Decreases* | | | | | | | |
| 78 | Thalamus R | 72 | Caudate R | −0.088 | 0 | 1 | 0 |
| 34 | Cingulum Mid R | 20 | Supp Motor Area R | −0.040 | 0 | 1 | 0 |
| 72 | Caudate R | 4 | Frontal Sup R | −0.028 | 0 | 1 | 0 |
| 34 | Cingulum Mid R | 32 | Cingulum Ant R | −0.026 | 0 | 1 | 0 |
| 72 | Caudate R | 24 | Frontal Sup Medial R | −0.022 | 0 | 1 | 0 |
| 71 | Caudate L | 3 | Frontal Sup L | −0.016 | 1 | 0 | 0 |
| 73 | Putamen L | 7 | Frontal Mid L | −0.014 | 1 | 0 | 0 |
| 24 | Frontal Sup Medial R | 3 | Frontal Sup L | −0.011 | 0 | 0 | 1 |
| 77 | Thalamus L | 7 | Frontal Mid L | −0.010 | 1 | 0 | 0 |

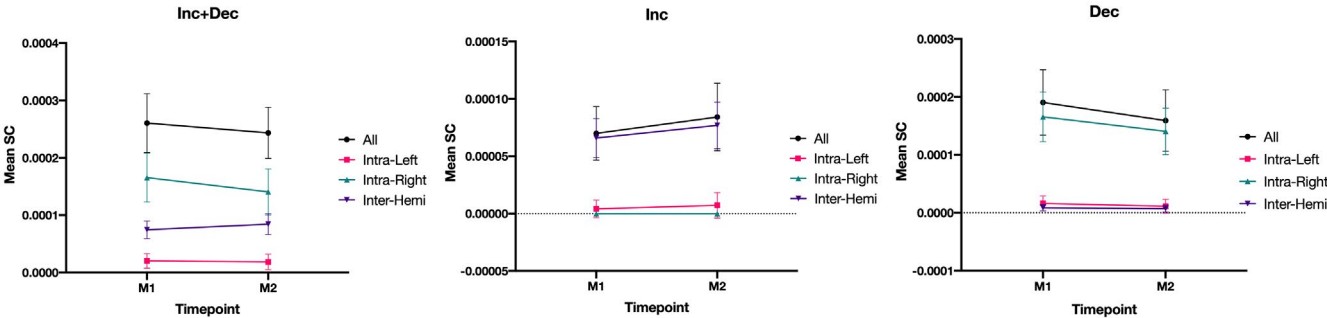

**FIGURE 2** Mean connectivity values of the significant connected component. (a) All the connections; (b) connections with increases in connectivity along time; (c) connections with decreases in connectivity. For each plot, we present the connectivity values for all connections (black), intra-left (red), intra-right (green), and inter-hemispheric (purple) connections. Intra-hemispheric connections exhibit a decrease along time, while inter-hemispheric links show an increase. Most of the decreases in SC are due to connections within the right hemisphere [Color figure can be viewed at wileyonlinelibrary.com]

## 3.5 | Topological organization longitudinal changes

### 3.5.1 | Graph theory metrics

Characteristic path length decreased significantly from the first to the second timepoint ($t_{(50)} = 3.45$, $p = 0.009$, $d = 0.29$). Regarding the other graph measures (node degree, connection density, global efficiency, local efficiency, clustering coefficient, and small-world index) no statistically significant differences were found (see Table S2).

### 3.5.2 | Hubs

Network hubs were defined as regions with high normalized nodal efficiency. In the first assessment, 13 regions were identified as hubs (Figures 5 and 6, Table 5). In timepoint 2, two hubs were lost in comparison to timepoint 1, that were left inferior parietal and left fusiform gyrus.

In the case of the provincial hubs, which play a central role in intra-modular communication, eight hubs were detected at timepoint

1 and 10 at timepoint 2. Only the right parahippocampal and bilateral fusiform gyrus were detected at both timepoints (Table 5).

Connector hubs represent a central role in inter-modular communication. At timepoint 1, five regions were identified as connector hubs, while at timepoint 2 only four regions were detected (Table 5). Left and right putamen were common to both timepoints.

### 3.5.3 | Modularity

The optimal modularity structure had six modules at both timepoints, and the two arrangements had a similarity of 0.80. Modules 4, 5, and 6 were common to both partitions. Module 3 changed from a leftward lateralization at timepoint 1 to a rightward lateralization at timepoint 2, which caused slight differences in the arrangement of the frontal regions of modules 1 and 2 (Figure 7a). Details of the

**TABLE 3** Percentage of longitudinal changes in the mean connectivity of the significant connected component. Percentages are given for all the connections comprising the connected component, only the connections with increases in connectivity and connections with decreases

| Network | All connections | Intra-left | Intra-right | Inter-hemispheric |
|---|---|---|---|---|
| All connections | −6.53 | −8.70 | −15.07 | 13.04 |
| Increases | 20.35 | 76.54 | 0 | 16.86 |
| Decreases | −16.40 | −30.10 | −15.07 | −15.9 |

regions belonging to each module are given in Table S3. Furthermore, analyzing the connectivity profile of the connector hubs (Figure 7b), we observe distinct patterns at the two timepoints. Even regions that were classified as connector hubs at both timepoints (left and right putamen), have different profiles of connectivity.

Regarding the overlap between modules and RSNs, despite their differences in the arrangement, the percentage of overlap has only very subtle differences in the first three modules. Module 1 has the highest overlap with both frontoparietal and somatomotor networks, with slightly higher overlap with frontoparietal network at timepoint 1. Module 2 again overlaps with frontoparietal (higher value at timepoint 2) and somatomotor, but also with the ventral attention network, and module 3 overlaps with limbic and default mode networks (DMNs) with higher value for the limbic network at both timepoints. In relation to the last three modules, module 4 overlaps with limbic and somatomotor networks, and modules 5 and 6 with visual and limbic networks. The matrices with the values of overlap at both timepoints are represented in Figure 7c.

### 3.5.4 | Fingerprints of modular connectivity

The reference scheme chosen to analyze fingerprints of modular connectivity was the community structure of timepoint 2. Significant alterations in connector-hub driven inter-modular connectivity were found (Figure 8). From timepoint 1 to timepoint 2, a decrease of around 19% of overall connectivity is found. Focusing on the specific

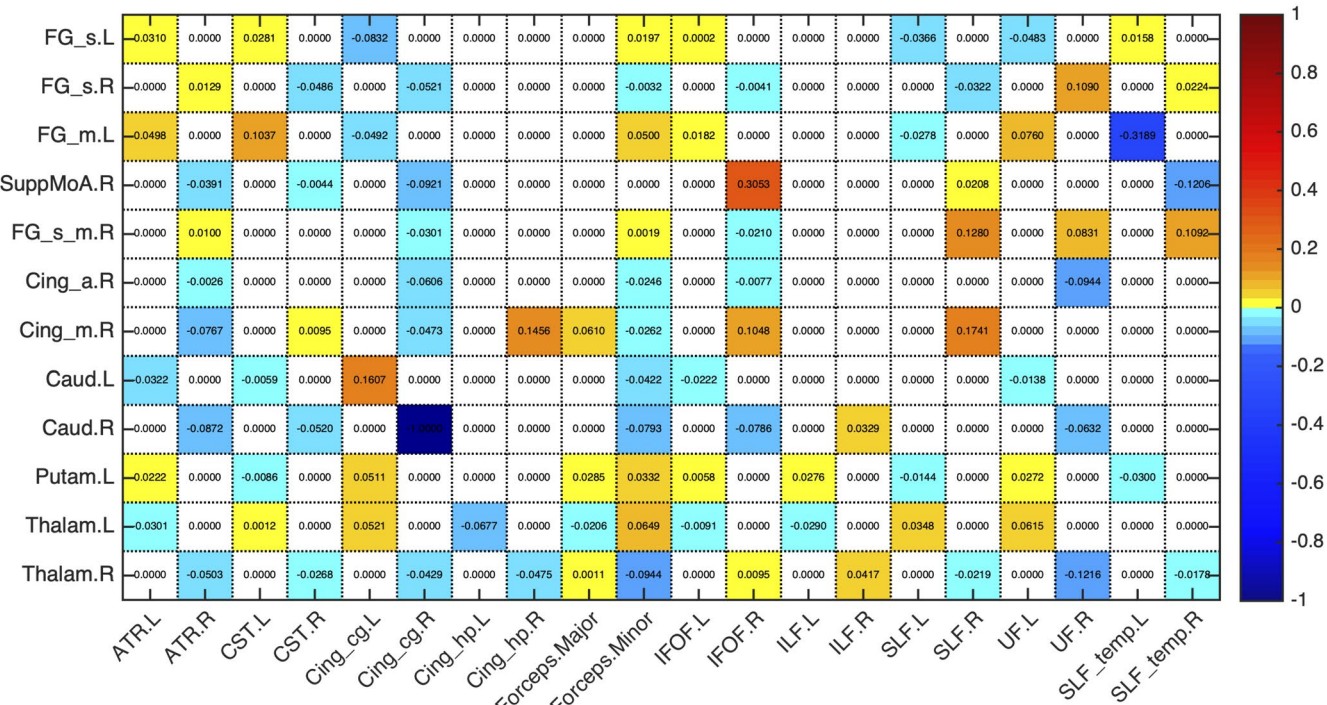

**FIGURE 3** Proportion of change between timepoints in the mean number of streamlines of the overlap between each seed region of the sub-network with decreases in structural connectivity and white matter (WM) tract. Seed regions are presented in rows and WM tracts in columns. For most of the connections, we found a common WM tract and the majority were association fibers [Color figure can be viewed at wileyonlinelibrary.com]

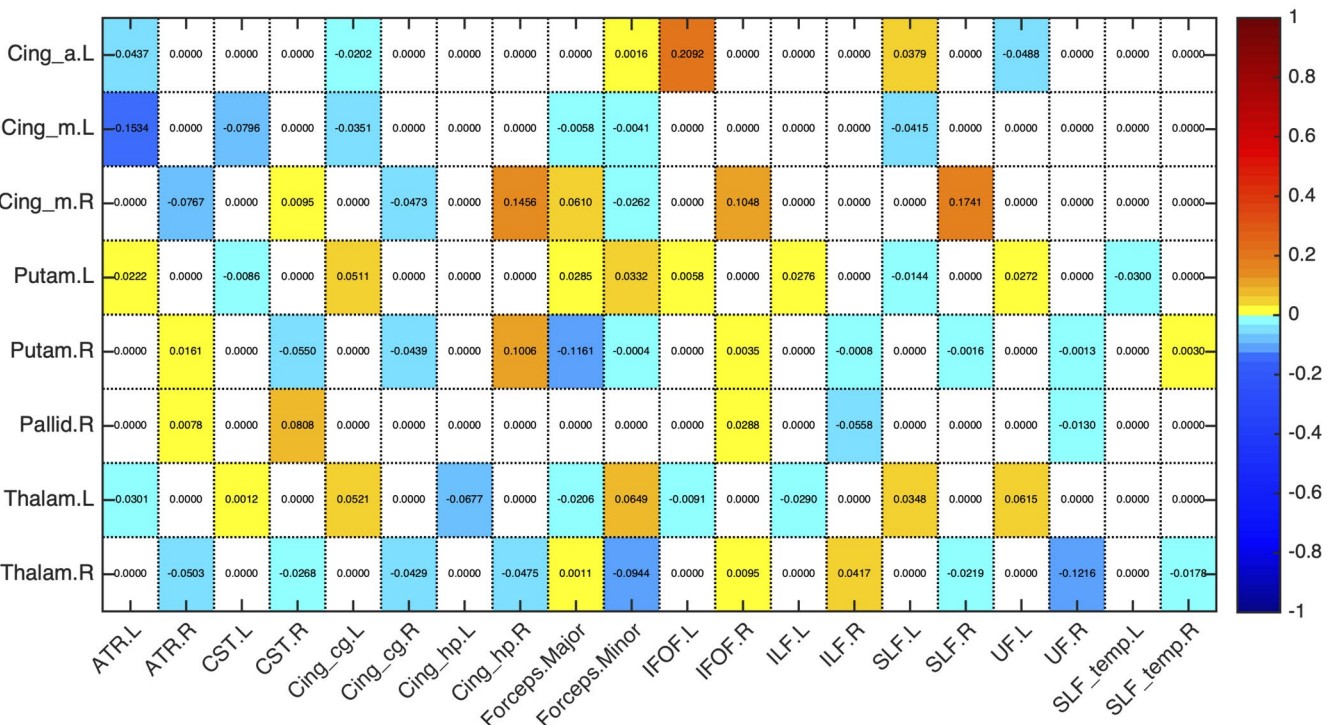

**FIGURE 4** Proportion of change between timepoints in the mean number of streamlines of the overlap between each seed region of the sub-network with increases in structural connectivity and white matter (WM) tract. Seed regions are presented in rows and WM tracts in columns. There was more than a single WM tract connecting the regions, probably due to the fact that almost all the connections were inter-hemispheric [Color figure can be viewed at wileyonlinelibrary.com]

connections that contribute to this decrease, at timepoint 1, we observe increased connectivity from module 2 (left hemisphere; frontal and parietal regions, insula, supramarginal, angular, putamen, pallidum, thalamus and superior temporal gyrus) to module 4 (bilateral supplementary motor area, middle and posterior cingulate cortex, precuneus and paracentral lobule) as well as from module 3 (frontal regions most on the right hemisphere, bilateral anterior cingulate cortex, caudate, right putamen, right pallidum, and right thalamus) to modules 2, 4, and 6 (left hemisphere; occipital and temporal regions, hippocampus, parahippocampal, amygdala, calcarine, cuneus, lingual and fusiform gyrus). Of notice, at timepoint 1 there are two connector hubs that belong to module 2 (left postcentral gyrus and left putamen) and also two connector hubs in module 3 (left caudate and right putamen), while at timepoint 2 there is only one connector hub in each of these modules (left putamen in module 2 and right putamen in module 3). Furthermore, modules 1 and 6, at timepoint 1, have no connectivity with any other modules, and the same happens with module 4 in timepoint 2 and module 5 at both timepoints. This is because there is a lack of connector hubs belonging to any of these modules, at the referred timepoints. No differences were found for intra-modular and inter-modular connectivity.

## 4 | DISCUSSION

In this study, we explored the longitudinal changes in the topological organization of brain structural networks in normal aging, using diffusion MRI. Our results revealed both decreases and increases in WM SC along time. Interestingly, the sub-network with decreasing connectivity is composed mainly of intra-hemispheric connections, while inter-hemispheric connections are in majority in the sub-network of increases. Both networks are mainly comprised by connections between subcortical and frontal regions. This differential pattern of changes in different types of connections could be explained by the last-in-first-out hypothesis, which claims that regions developing later are more prone to age-related decline (Raz, 1999). In terms of WM tracts, association fibers, that connect different regions within a hemisphere, have a later peak of maturation when compared to commissural fibers that connect regions between the two hemispheres (Hermoye et al., 2006). According to this hypothesis it is expected that association fibers will undergo a steeper decline in comparison to commissural fibers, which was already demonstrated in studies of WM microstructural properties indirectly estimated from DTI-based metrics (Bender et al., 2016; Brickman et al., 2012; Davis et al., 2009; Slater et al., 2019). Our results are in line with these findings, since intra-hemispheric links decline along aging, while inter-hemispheric links appear to be maintained and even enhanced. This enhance in connectivity between hemispheres along aging, has been reported in recent functional studies. One such study found stronger FC between bilateral frontoparietal control network that was associated with better cognition in the visuospatial domain (Jiang et al., 2020). Moreover, there are also reports of increased bilateral frontal activation in episodic memory retrieval tasks, for high-performing older adults (Cabeza

**TABLE 4** WM tracts connecting each pair of regions of the significant sub-networks with structural connectivity differences between timepoints

| Area 1 | | Area 2 | | WM Tract |
|---|---|---|---|---|
| N | Name | N | Name | |
| *Increases* | | | | |
| 77 | Thalamus L | 74 | Putamen R | CST L; IFOF R |
| 78 | Thalamus R | 73 | Putamen L | IFOF R; IFOF L |
| 76 | Pallidum R | 73 | Putamen L | IFOF R; IFOF L |
| 77 | Thalamus L | 33 | Cingulum Mid L | CST L; CGC L |
| 73 | Putamen L | 31 | Cingulum Ant L | IFOF L; FMI |
| 78 | Thalamus R | 77 | Thalamus L | IFOF R; CST L |
| 34 | Cingulum Mid R | 33 | Cingulum Mid L | CST R; CST L |
| *Decreases* | | | | |
| 78 | Thalamus R | 72 | Caudate R | ATR R |
| 34 | Cingulum Mid R | 20 | Supp Motor Area R | CGC R |
| 72 | Caudate R | 4 | Frontal Sup R | FMI |
| 34 | Cingulum Mid R | 32 | Cingulum Ant R | CGC R |
| 72 | Caudate R | 24 | Frontal Sup Medial R | CGC R |
| 71 | Caudate L | 3 | Frontal Sup L | UF L |
| 73 | Putamen L | 7 | Frontal Mid L | SLF L |
| 24 | Frontal Sup Medial R | 3 | Frontal Sup L | CGC R; CGC L |
| 77 | Thalamus L | 7 | Frontal Mid L | ATR L; SLF L |

Abbreviations: ATR, anterior thalamic radiation; CGC, cingulate gyrus part of cingulum; CST, corticospinal tract; FMI, forceps minor; IFOF, inferior fronto-occipital fasciculus; SLF, superior longitudinal fasciculus; UF, uncinate fasciculus.

et al., 2002). Another study found that higher cognitive status in healthy older adults was associated with higher between-network and inter-hemispheric FC (Sullivan et al., 2019). Our findings add to this evidence by demonstrating that this increase in bilateral connectivity also occurs in SC. In terms of WM SC, there are few reports showing age-related increases. Of those, Lee and colleagues (Lee et al., 2015) found age-related increases between the prefrontal cortex and temporal regions, and between occipital and posterior brain regions. In our study, we found age-related increases between frontal and subcortical regions.

Our analysis of the WM tracts involved in the connections of each sub-network further supports the last-in-first-out hypothesis. The sub-network with decreases in SC was mainly composed of association fibers, with only one commissural fiber (forceps minor). While the sub-network with increases in SC, although it also included association fibers, was comprised by many commissural and projection fibers. It should be noted that connections with increasing connectivity are mainly inter-hemispheric, so it is very probable that the corpus callosum is involved in all these connections. Association fibers are the latest to develop in comparison to commissural and projection fibers and previous studies have reported

more pronounced decline of DTI-based metrics for association fibers, which may possibly indicate a steepest decline in WM integrity (Bender et al., 2016; Benitez et al., 2018; Bennett & Madden, 2014; Cox et al., 2016; de Groot et al., 2015). Our results conform with these findings by demonstrating that disruption in WM SC occurs primarily in association fibers.

Furthermore, we analyzed the association between the mean connectivity of these sub-networks and cognitive scores of memory (MEM) and global cognition and executive function (EXEC). We found a trend in the correlation between the sub-network of increases and EXEC. Specifically, higher SC values were associated with lower EXEC scores. Previous studies reported that age-related FC increase was associated with poorer cognitive performance (Chen et al., 2019; Nashiro et al., 2017). Our finding is in line with the dedifferentiation hypothesis of the aging brain, which suggests that age is accompanied by a loss of specificity in the neural response to cognitive tasks (Chan et al., 2014; Dennis & Cabeza, 2011; Geerligs et al., 2014, 2015; Goh, 2011; Park et al., 2004). Our result on SC might suggest that the increase in SC is necessary to recruit additional areas in order to try to compensate for the cognitive decline these older adults are experiencing. Although we still see a decline in cognitive performance, this increase in SC is probably critical for the older adults to be able to perform cognitive tasks. Further research will be needed to confirm that this association reflects a compensatory mechanism.

Analysis of the topological features of brain WM structural networks revealed some, although few, longitudinal alterations. No significant differences were found in most of the analyzed graph metrics, namely, node degree, connection density, global and local efficiency. Although some studies report age-related declines in some of these metrics, others present null results. Regarding global efficiency, (Wen et al., 2011; Zhao et al., 2015) report reduced global efficiency in advanced ages, whereas (Gong et al., 2009) found no significant age effect on this metric. Thus, the existence of controversial results might explain the lack of significant results in our study. Moreover, the limited sample size used in this study is a limitation which may have contributed to the lack of significant alterations in these measures. Characteristic path length was the only metric found to be significantly different between timepoints, having lower values in the last timepoint. This finding means that the average shortest path length between all possible pairs of nodes in the network was lower in the last assessment and thus, globally, the communication between different regions was more efficient. This result is not in accordance with earlier studies which reported increases in characteristic path length (Fischer et al., 2014; Zhu et al., 2012), but it should be noted that the effect size for this significant difference was rather small ($d = 0.29$) and this metric is inversely related to global efficiency, in which we found no significant differences.

In nodal efficiency, differences were found between assessments, with the loss of two hubs (left inferior parietal and left fusiform gyrus) from the first to last timepoint. Left inferior parietal cortex (IPC) is known to be associated with language processing, namely, reading, phonology, and semantic processing (Amici

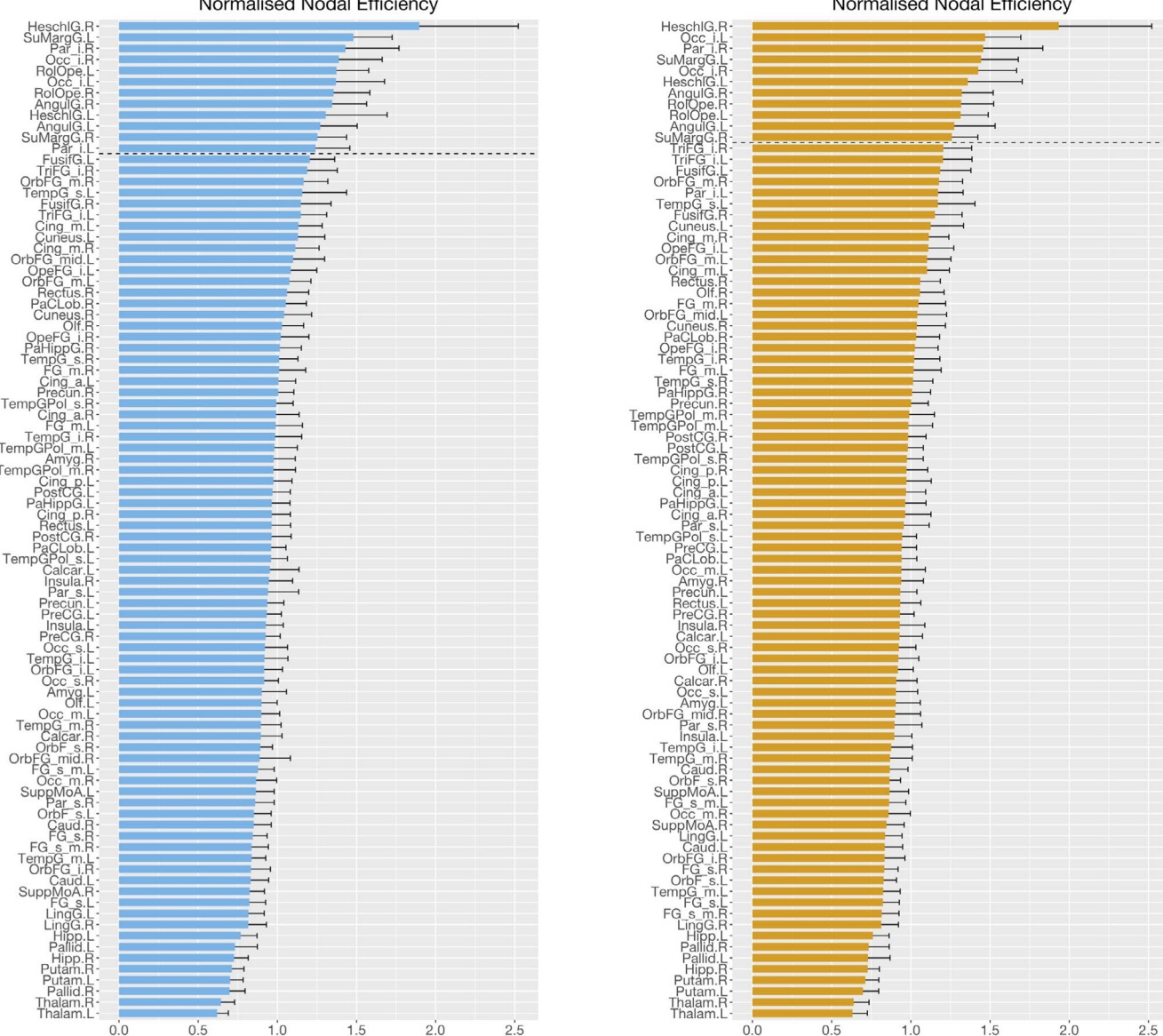

**FIGURE 5** Global hubs identified in the two timepoints as measured by the normalized nodal efficiency. Here, we observe the plot of the normalized nodal efficiency for all the 90 automated anatomical labeling (AAL) regions, sorted in descending order of efficiency values, for timepoint 1 (left) and timepoint 2 (right). We observe a reorganization of brain structural networks in aging, characterized by the loss of two hubs (left inferior parietal cortex and left fusiform gyrus) [Color figure can be viewed at wileyonlinelibrary.com]

et al., 2006; Gorno-Tempini et al., 2004; Graves et al., 2010; Price & Mechelli, 2005; Turkeltaub & Branch Coslett, 2010; Vigneau et al., 2006). Additionally, it also plays a significant role in episodic memory (Wagner et al., 2005), attention (Corbetta & Shulman, 2002; Fan et al., 2005), action and salience processing (Behrmann et al., 2004; Caspers et al., 2010; Iacoboni, 2005), and social cognition (Bzdok et al., 2016). It is also an important node of the DMN (Greicius et al., 2003). Previous aging studies have found age-related changes in the FC of this region and an association between these changes and cognitive decline in different domains, specifically, executive function (Lou et al., 2019; Zhao et al., 2020), semantic knowledge (Hoffman & Morcom, 2018), inhibitory control (Hu et al., 2018),

and memory (Huo et al., 2018; Lamichhane et al., 2018). In line with our results, there is evidence of the loss of hub role for left IPC in aging, which was demonstrated using cortical thickness covariance networks (Carey et al., 2019). Regarding fusiform gyrus, this region is involved in object recognition (Grill-Spector et al., 2001), face perception (Kanwisher et al., 1997), including haptic and visual identification of faces (Kitada et al., 2009), reading (Cohen et al., 2000; Wandell et al., 2007) and memory (Wagner et al., 1999). Functional studies of aging using face recognition tasks have demonstrated a relationship between patterns of activation in the fusiform gyrus and age-related declines in face recognition or perception (Dennis et al., 2008; Lee et al., 2011; Wright et al., 2008). There is also

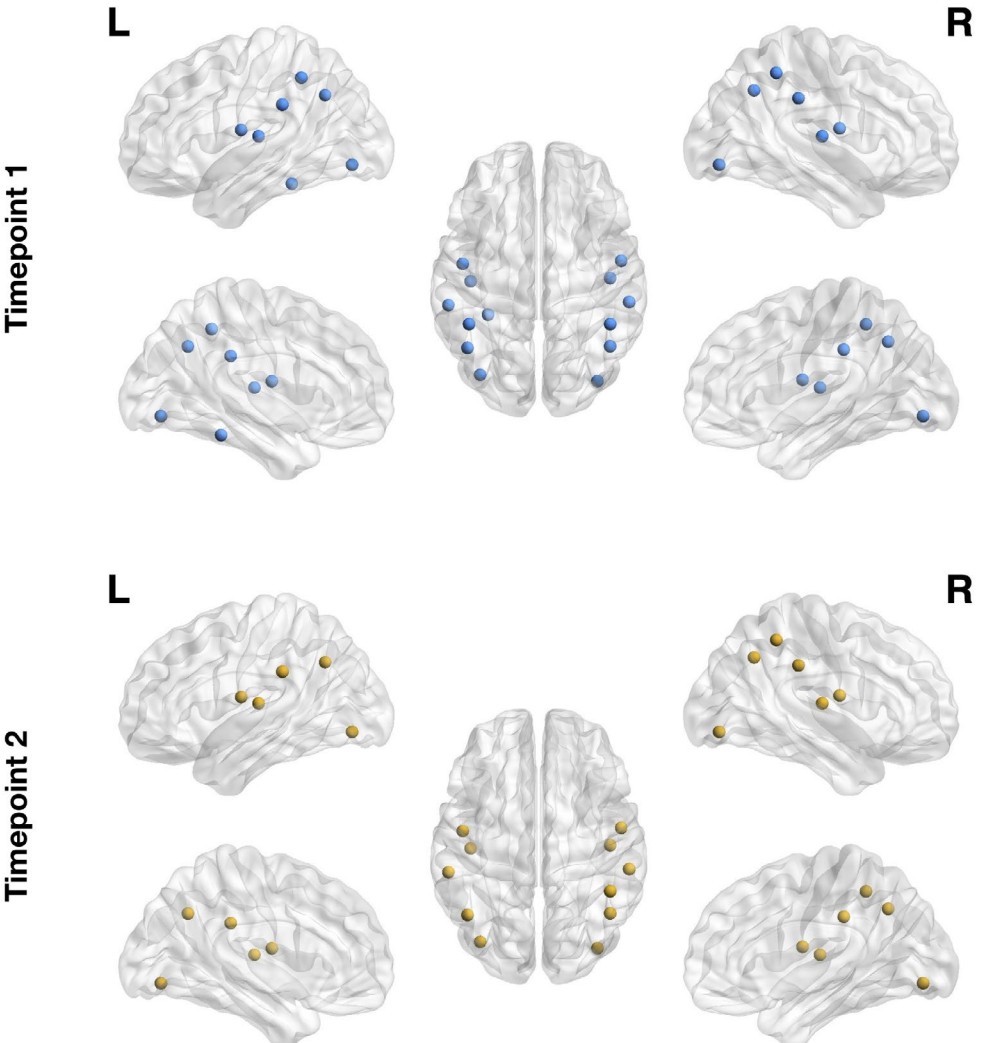

**FIGURE 6** Global hubs identified in the two timepoints as measured by the normalized nodal efficiency. Here, we represent in the brain the identified hubs for timepoint 1 (top row) and timepoint 2 (bottom row). We observe a reorganization of brain structural networks in aging, characterized by the loss of two hubs (left inferior parietal cortex and left fusiform gyrus) [Color figure can be viewed at wileyonlinelibrary.com]

evidence of age-related atrophy of the fusiform gyrus (Hogstrom et al., 2013; Shah et al., 2020). Supporting our results, previous studies found the loss of hub role for the left fusiform characterized by reduced betweenness centrality in cortical networks of regional gray matter volumes (Li et al., 2018).

Concerning modularity structure, we found very similar modular arrangements at both timepoints, with only some differences in frontal regions. Regarding fingerprints of modular connectivity, longitudinal differences were found in connector-hub driven inter-modular connectivity. Overall, there was a decrease of around 19% in this connectivity from timepoint 1 to timepoint 2, which could be the result of the loss of one connector hub. This decrease suggests a reduction of integration of brain structural networks during aging, since connector hubs play an important role in inter-modular communication (characterized by high participation coefficient) and thus, there is less communication between different functional modules of the brain. There is already evidence suggesting a decrease in

integration and increase in segregation of brain functional networks during the aging process (Sala-Llonch et al., 2014).

Between assessments, two regions lost their connector hub role, namely left caudate and right midcingulate cortex, while left middle occipital gyrus was identified as a connector hub only at the last timepoint. Both left caudate and right midcingulate cortex were part of the identified sub-network with decreases in connectivity between timepoints. Thus, this decrease in the anatomical connections between either left caudate or right midcingulate cortex and other regions of the brain would be expected to isolate these two regions and could explain the loss of these two nodes as connector hubs. Caudate is associated with different aspects of cognition, including motor and action planning, decision-making (particularly, goal-directed behavior), motivation and reward processing (Bick et al., 2019; Cera et al., 2019; Grahn et al., 2008; Wilson, 2018). Previous studies exploring age-related changes in the caudate have shown associations with different cognitive domains, such

**TABLE 5** Hubs of the brain for the two timepoints, according to three classification methods used. Global hubs are sorted by nodal efficiency, and provincial and connector hubs are sorted by modularity degree z-score

| Global hubs | | Provincial hubs | | Connector hubs | |
| --- | --- | --- | --- | --- | --- |
| M1 | M2 | M1 | M2 | M1 | M2 |
| Heschl R | Heschl R | Frontal Sup Orb L | Rolandic Operculum R | Cingulum Mid R | Occipital Mid L |
| SupraMarginal L | Occipital Inf L | ParaHippocampal R | ParaHippocampal R | Postcentral Gyrus L | Putamen R |
| Parietal Inf R | Parietal Inf R | Insula R | Temporal Inf L | Caudate L | Postcentral Gyrus R |
| Occipital Inf R | SupraMarginal L | Parietal Inf L | Rectus R | Putamen L | Putamen L |
| Rolandic Operculum L | Occipital Inf R | Rolandic Operculum L | Parietal Inf R | Putamen R | |
| Occipital Inf L | Heschl L | Rectus L | Frontal Med Orb R | | |
| Rolandic Operculum R | Angular R | Fusiform L | Caudate R | | |
| Angular R | Rolandic Operculum R | Fusiform R | Fusiform L | | |
| Heschl L | Rolandic Operculum L | | Fusiform R | | |
| Angular L | Angular L | | Insula L | | |
| SupraMarginal R | SupraMarginal R | | | | |
| Parietal Inf L | | | | | |
| Fusiform L | | | | | |

Abbreviations: M1, timepoint 1; M2, timepoint 2.

as episodic memory (Fjell, Sneve, Storsve, et al., 2016; Rieckmann et al., 2018), instrumental learning (Perosa et al., 2020), cognitive flexibility (Verstynen et al., 2012), and reward processing (Bowen et al., 2020; Dhingra et al., 2020). Interestingly, (Esteves et al., 2018) reported that older adults had overall longitudinal rightward lateralization of the caudate volume and subjects with extreme increase in this rightward asymmetry had increased Stroop interference scores (i.e., a measure of cognitive flexibility) but decreased scores of general cognition. Our results provide additional support to this, by showing that the left caudate also loses its role of integrating different regions of the brain. Midcingulate cortex is associated with motor control (from self-initiated movements to reflexive motor activity), and also with the response to acute nociceptive stimuli, fear, and pain (Hoffstaedter et al., 2015; Vogt, 2016). A previous study found age-related reductions in FC between dorsal anterior insula and midcingulate cortex, which are part of the dorsal salience sub-network and these changes were found to be a mediator of age-related declines in executive function (Touroutoglou et al., 2018). Another study elucidated the role of midcingulate cortex in motor functions. Specifically, this region is involved in a network associated with intentional movement initiation and it was found to present decreased FC with anterior cingulate motor area in aging, and there was also a decrease in gray matter volume with age (Hoffstaedter et al., 2015). These previous studies suggesting decreased connectivity and atrophy of this region with aging could explain the loss of connector hub status of the midcingulate cortex in our results.

At the last timepoint, left middle occipital gyrus emerged as a connector hub. This region is associated with visual information processing and communication (Anurova et al., 2015; Teng et al., 2018; Wandell et al., 2007), and also plays a role in the perception of facial emotion as well as in category-selective attention modulating unconscious face/tool processing (Tu et al., 2013). Previous studies found age-related differences in the patterns of activation of the middle occipital gyrus during visual tasks (Berghuis et al., 2019; Piefke et al., 2012). Additionally, in a study of autobiographical memory retrieval, increases in activation of middle occipital gyrus in older adults were found during episodic memory retrieval, which could reflect a compensatory mechanism due to impairment of vivid visual imagery, or higher use of visuospatial processing during episodic memory retrieval (Donix et al., 2010). This compensatory increase in activation of middle occipital gyrus might explain its appearance as a connector hub at the last timepoint.

Lastly, there was an increase in detected provincial hubs along time, which may reflect higher segregation/specialization of structural networks. These hubs are characterized by having most of their connections within their own module and thus play a key role in intra-modular communication. The additional regions detected at timepoint 2 were right caudate and left inferior temporal gyrus. As described before, caudate was found to have a rightward asymmetry in aging, what may explain the gain of provincial hub status along time. Inferior temporal gyrus is associated with semantic processing, particularly the selection and controlled retrieval of information from memory (Thompson-Schill, 2003), and it has also been involved in intelligence and executive function (Jung & Haier, 2007). Some studies report relative preservation of cortical thickness of this region until later in life (Fjell et al., 2009; Lee et al., 2018), and this can be related to the preservation of semantic memory also observed in aging (Gold et al., 2009). Our results also support the maintenance of

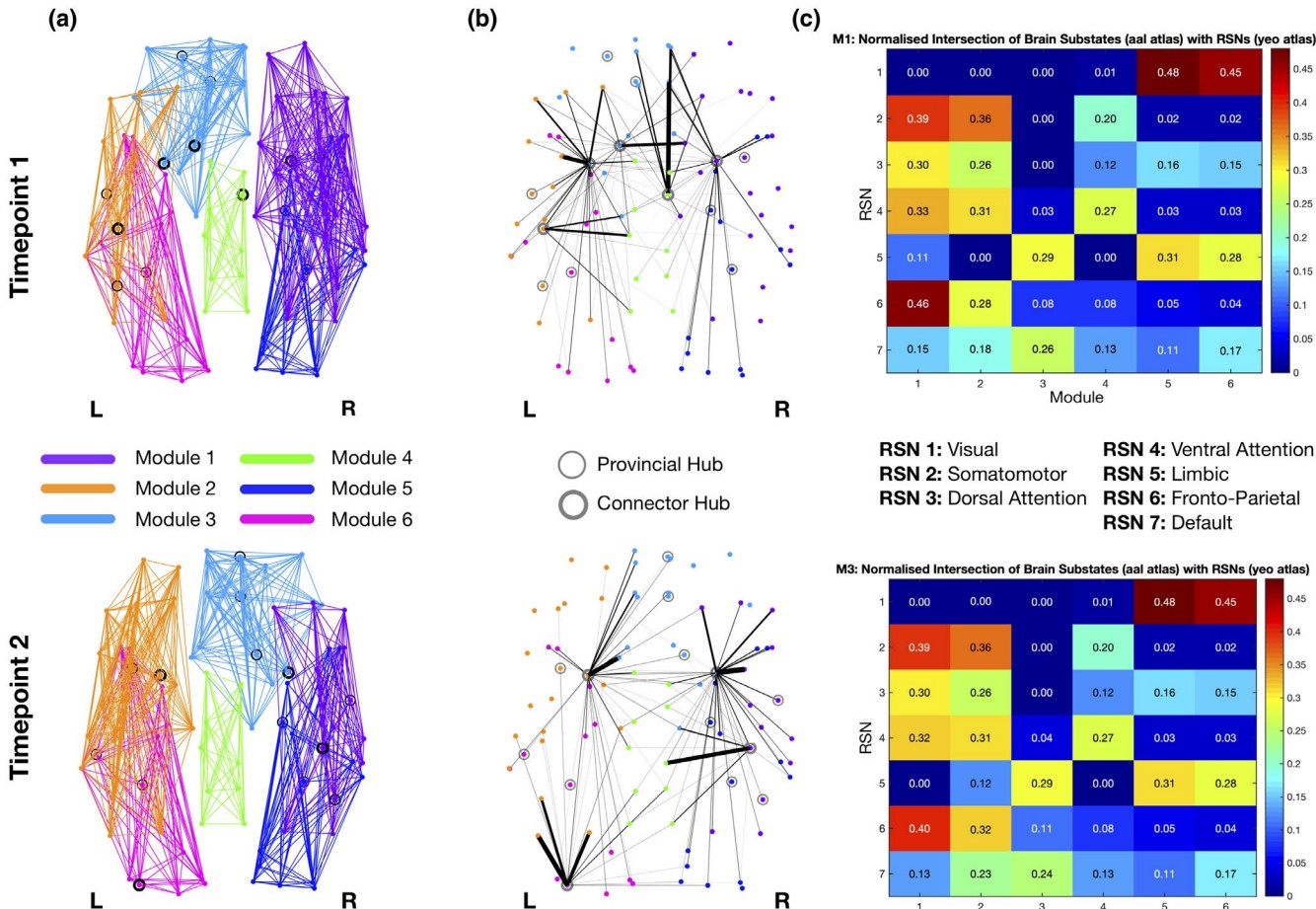

**FIGURE 7** Modularity structure (a), connector-hub connectivity (b) and matrices of resting-state networks (RSNs) overlap (c) at timepoint 1 (top row) and timepoint 2 (bottom row). Filled circles represent connector hubs and unfilled circles represent provincial hubs. Although very similar modular arrangements were found at both timepoints (a,b), the undirected structural connectivity profile for the connector hubs was different (c). These differences are probably due to the loss of two connector hubs from first to last timepoint, namely left caudate and right midcingulate cortex, while left middle occipital gyrus was identified as a connector hub only in the last timepoint. Giving the role of connector hubs in inter-modular communication, the reduction in their number between timepoints reflects a decrease in integration of brain structural networks in aging [Color figure can be viewed at wileyonlinelibrary.com]

this region along aging as it was characterized as an important hub of the brain network in the last timepoint.

This study has some limitations, particularly the low sample size and the period between evaluations. This could explain why there were almost no differences in graph theory metrics, and thus an extended period of evaluation and a larger sample could allow to observe differences in these measures. Another limitation is the use of a 1.5T MRI scanner which has lower signal to noise ratio when compared to 3T MRI scanners (Lee & Shannon, 2007). Future studies will benefit from using a 3T scanner, which will allow to obtain high-quality images. While these limitations may have had some influence on the obtained results (changes in network connectivity, hubs, and modularity structure), we believe these effects were minimal since the same protocol (same scanner, acquisition parameters, and data processing pipeline) was used at both evaluations.

In summary, our findings bring further support of the existing evidence of the reorganization of brain structural networks during aging. Specifically, we found decreases in intra-hemispheric

connectivity and increases in inter-hemispheric connectivity. Association fibers were primarily responsible for the decreases in WM SC and their functional loss is consistent with the last-in-first-out hypothesis. Additionally, we found a trend for an association between cognition and a sub-network with increasing connectivity, exhibiting lower general cognition and executive functioning scores for higher SC values, possibly suggesting some form of a compensatory mechanism. Regarding topological features of brain networks, we found evidence suggesting reduced integration, characterized by a decrease in connector-hub driven inter-modular connectivity, and increased segregation, portrayed as an increase in the number of detected provincial hubs, of brain structural networks in aging. Taken together, these findings elucidate the changes occurring in the brain during aging, in terms of communication between the hemispheres and between specialized modules. This can help identify brain regions responsible for this disruption, that could be targeted as biomarkers to prevent cognitive decline in aging.

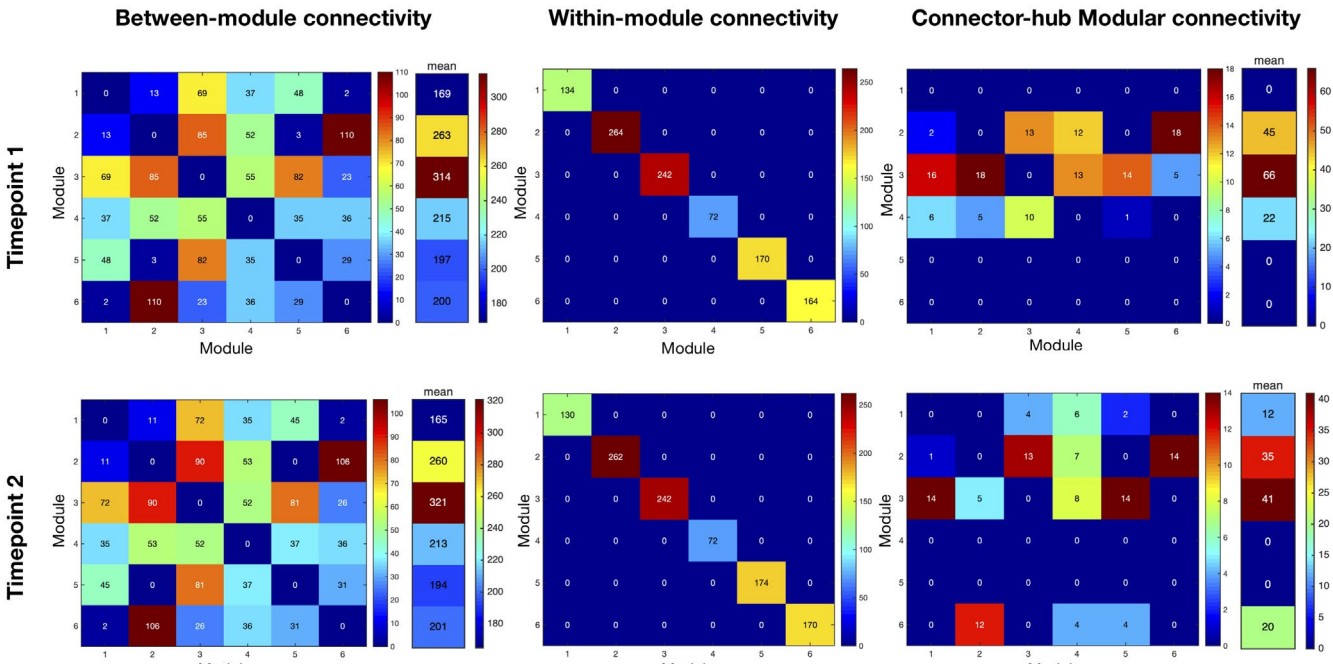

**FIGURE 8** Fingerprints of modular connectivity at timepoint 1 (top row) and timepoint 2 (bottom row). Left column represents the inter-modular connectivity, middle column the intra-module connectivity, and right column the connector-hub driven inter-modular connectivity. Modular connectivity strength is quantified as the total number of connections (degree) of all nodes forming a module. Community structure of timepoint 2 was selected as the reference scheme, since it had higher group goodness-of-fit. We observe different patterns only in connector-hub driven inter-modular connectivity. Overall, there was a decrease of around 19% in this connectivity between timepoints, which is probably due to the loss of one connector hub. These results again suggest a decrease in integration of brain structural connectivity (SC) during aging [Color figure can be viewed at wileyonlinelibrary.com]

## 5 | COMPLIANCE WITH ETHICAL STANDARDS

All procedures followed were in accordance with the ethical standards of the responsible committee on human experimentation (institutional and national) and with the Helsinki Declaration of 1975, and the applicable revisions at the time of the investigation. Informed consent was obtained from all patients for being included in the study.

## DECLARATION OF TRANSPARENCY

The authors, reviewers and editors affirm that in accordance to the policies set by the *Journal of Neuroscience Research*, this manuscript presents an accurate and transparent account of the study being reported and that all critical details describing the methods and results are present.

## ACKNOWLEDGMENTS

The authors thank the study participants.

## CONFLICT OF INTEREST

The authors declare that they have no conflict of interest.

## AUTHOR CONTRIBUTIONS

*Conceptualization*, N.S. and N.C.S.; *Methodology*, N.S. and N.C.S.; *Resources*, L.A., T.C. and C.P.-N.; *Investigation*, L.A., T.C., J.S.M., R.M., P.S.M., P.M. and A.C.; *Formal Analysis*, A.C. and H.M.F.; *Software*, A.C. and H.M.F.; *Writing - Original Draft*, A.C.; *Writing - Review & Editing*, A.C., H.M.F., R.M., P.S.M., P.M., J.M.S., L.A., C.P.-N., T.C., N.C.S. and N.S.; *Visualization*, A.C.; *Supervision*, N.S. and N.C.S.; *Funding Acquisition*, N.S. and N.C.S.

## PEER REVIEW

The peer review history for this article is available at https://publons.com/publon/10.1002/jnr.24795.

## DATA AVAILABILITY STATEMENT

The data sets generated during and/or analyzed during the current study are available from the corresponding author on reasonable request.

## ORCID

*Ana Coelho* https://orcid.org/0000-0001-8489-5750

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

## SUPPORTING INFORMATION

Additional supporting information may be found online in the Supporting Information section.

Transparent Science Questionnaire for Authors

Transparent Peer Review Report

**FIGURE S1** Correlation between head-motion relative displacement values and age for all subjects and both timepoints. Head-motion displacement values were extracted using FSL tools and averaged across all volumes acquired for one subject. Correlation is not significant ($r = 0.019$, $p = 0.85$) meaning that age is not associated with head-motion

**FIGURE S2** Comparison of head-motion relative displacement values between timepoints. A paired *t*-test was performed, and it was not significant ($p = 0.95$) meaning that head-motion values did not differ between timepoints

**FIGURE S3** Percentage of connections lost in each subject when applying consistency-based thresholding. Percentage is calculated as the proportion of connections removed in the subject SC matrix relative to the total number of connections removed in the group consistency mask. Plot on the left illustrates results for timepoint 1 and on the right, results for timepoint 2

**FIGURE S4** Frequency distribution for the connection strength of the links removed when applying consistency-based thresholding. Plot on the left illustrates results for timepoint 1 and on the right, results for timepoint 3

**FIGURE S5** Percentage of connections that were present in the group consistency mask but were not present in all subjects' SC matrices. Percentage is calculated as the proportion of connections not present in the subject SC matrix relative to the total number of connections in the group consistency mask. Plot on the left illustrates results for timepoint 1 and on the right, results for timepoint 2

**FIGURE S6** Frequency distribution for the connection strength of the links from the group consistency mask not present in all subjects, when applying consistency-based thresholding. Plot on the left illustrates results for timepoint 1 and on the right, results for timepoint 3

**FIGURE S7** Consistent signatures of SC for M1 and M2 timepoints. Left panel shows intra-timepoint consistency measured as the association between individual SC signatures and timepoint average SC and we can observe that the two timepoints reveal a very high level of intra-timepoint consistency (M1: 97.6%; M2: 97.5%). Right panel shows the degree of association between the signatures of SC for all pairs of subjects in the same timepoint. Once again, we notice a high level of timepoint consistency in SC (100% and 99.8% of all pairwise combinations in M1 and M2 timepoints respectively have a correlation higher than $r = 0.9137$, with number of occurrences peaking at approximately $r = 0.96$ for both timepoints). The overlap between the distributions of intra-timepoint consistency of both timepoints is additionally confirmed by the inter-timepoint consistency distribution (M1-M2: peak at approximately $r = 0.95$). Taken together, these results suggest that, at a global level, the patterns of SC are highly consistent within and between timepoints, and thus potential differences due to age and sex do not have a significant impact on the estimation of SC patterns

**FIGURE S8** Relationship between F-threshold and number of connections/nodes, that detected a significant component. The F-threshold used in this study (17.0) was selected based on the maximal F-threshold that detected a single component with more than two connections. This generated an NBS component with 19% nodes of the network and 16 links

**FIGURE S9** Repeated measures correlation between mean SC values of the network with increases and mean factor scores of general cognition and executive function

**FIGURE S10** Values of the mean number of streamlines for seed regions of the sub-network with decreases in structural connectivity. Top row shows values for timepoint M1 and bottom row shows values for timepoint M2. Seed regions are presented in rows and white matter tracts in columns

**FIGURE S11** Values of the mean number of streamlines for seed regions of the sub-network with increases in structural connectivity. Top row shows values for timepoint M1 and bottom row shows values for timepoint M2. Seed regions are presented in rows and white matter tracts in columns

**TABLE S1** Correlations between mean SC values of sub-networks and cognitive composite dimensions (MEM and EXEC)

**TABLE S2** Timepoint differences in graph theory metrics (results FDR corrected at $p < 0.05$)

**TABLE S3** Brain regions belonging to the different modules of each timepoint's modularity community structure

