## [Transparent Peer Review Report · Journal of Neuroscience Research]

Reorganization of brain structural networks in aging: a longitudinal study

**Coelho, Ana; Fernandes, Henrique; Magalhães, Ricardo; Moreira, Pedro; Marques, Paulo; Soares, José;
Amorim, Liliana; Portugal-Nunes, Carlos; Castanho, Teresa; Santos, Nadine; Sousa, Nuno**

Review timeline:

Submission date: 15 Jul 2020

Editorial Decision: Major Modification (12-Sep-2020)

Revision Received: 2 Nov 2020

Editorial Decision: Minor modification (16-Dec-2020)

Revision Received: 16 Dec 2020

Accepted: 31 Dec 2020

Editor 1: Jeremy Hogeveen
Editor 2: Junie Warrington
Reviewer 1: Jenny Reick
Reviewer 2: Andrew Bender

1st Editorial Decision

Decision letter

Dear Dr

Coelho:

Thank you for submitting your manuscript to the Journal of Neuroscience Research. We have now received the reviewer feedback and have appended those reviews below. As you will see, the reviewers find the question addressed to be of potential interest. Yet, they do not find the manuscript suitable for publication in its current form.

If you feel that you can adequately address the concerns of the reviewers, you may revise and resubmit your paper within 90 days. It will require further review. Please explain in your cover letter how you have changed the present version. If you require longer than 90 days to make the revisions, please contact Dr Junie Warrington (jpwarrington@umc.edu). You can submit your revised manuscript directly by clicking on the following link: *** PLEASE NOTE: This is a two-step process. After clicking on the link, you will be directed to a webpage to confirm. ***

https://mc.manuscriptcentral.com/jnr?URL_MASK=9f3c47d1a240496b915dcfd6b569780f

Thank you again for your submission to the Journal of Neuroscience Research; we look forward to reading your revised manuscript.

Best Wishes,

Dr Jeremy Hogeveen
Associate Editor, Journal of Neuroscience Research

Dr Junie Warrington
Editor-in-Chief, Journal of Neuroscience Research

Associate Editor: Hogeveen, Jeremy

Comments to the Author:

Dear Dr. Sousa,

Thank you for submitting your work to the Journal of Neuroscience Research. The manuscript has now been reviewed by myself and two external expert reviewers. While the reviewers and I agree that the longitudinal approach taken in this study represents an important advance given most prior work has been cross-sectional, there are significant concerns with a lack of clarity regarding key methodological details, and insufficient integration of relevant prior studies into the Discussion. If you are able to address the reviewers' significant concerns, I would invite you to revise and resubmit this manuscript to JNR.

Best Wishes,

Dr Jeremy Hogeveen
Associate Editor, Journal of Neuroscience Research

Reviewer: 1

Comments to the Author

The current study examines structural connectome metrics (generated from diffusion weighted imaging) in a longitudinal sample of older adults and reports changes in intra-hemispheric connectivity as well as increased segregation over time. The longitudinal nature of this work makes it an important addition to the literature which largely has examined age differences cross-sectionally. My questions and comments largely regard clarification of some of the methods and interpretation of these findings as well as a suggestion to control for potential additional confounds (ie head motion)

1. Methods: Was head-motion during the diffusion scan accounted for in any of the analyses? Prior work has shown (particularly in aging populations that tend to move more in scanner) that increased head motion can selectively different connectome estimates (eg short vs long connection), even after correcting for motion during preprocessing (see Baum et al., 2018, Neuroimage). Do the results hold when statically controlling for motion (ie including frame displacement as a nuisance covariate) in the analyses?
2. Methods: For the PCA of the neurocognitive measures, why did the authors include MMSE as a measure (as it is generally considered a more clinical/screening assessment)? Was the total MMSE score used (summed across all questions)? Because the MMSE includes sections that specifically test memory vs more executive functions, it is not clear how total MMSE score would fit into either component of the PCA, so if the authors believe MMSE should be included, it would make more sense to include sub-section scores rather than the total score.
3. Methods: The PCA of the neurocognitive measures is not clear – were two separate PCAs conducted per time-point or were data pooled together? The authors also describe z-scored cognitive measures, but it is also not clear how they were generated (z-scoring of the PCA factor scores?). They report mean z-scores that are not equal to 0 (as would be expected from the z-scoring process) which also has me confused about how this was conducted.
4. Discussion: The authors largely describe their structural connectome findings in relation to functional connectivity literature, but it would be helpful to also relate their findings to age-effects white matter structure. Do the connections they find map onto major white matter bundles? I find it curious that the authors do not use the term “white matter” at all in the discussion, so are these structural connectivity metrics not meant to represent properties of underlying white matter structure?
5. Discussion: Similar to the above point, what are the neurobiological changes that might account for the reported reorganization of structural connectivity (ie the finding of increased connectivity estimates)? Are new white matter connections being made? Is myelination along existing connections increasing? Again, the authors never explicitly mention “white matter” in the discussion so it is not entirely clear to me what “structural connectivity” is truly representing at the neurological level.
6. Discussion: The authors describe the finding of increased connectivity being associated with worse cognition as compensatory. However, recent guidelines put out by many leaders in the field of neurocognitive aging stress that the term compensation should be used to refer to cognition-enhancing brain differences (Cabeza et al., 2019, Nat Rev Neurosci). The authors should reassess how their findings fit in with the terminology put forth in the recent guidelines.

Reviewer: 2

Comments to the Author

The manuscript reports efforts to characterize longitudinal changes in graph theory-based structural brain network properties derived from diffusion MRI data in middle- and older aged adults. This effort represents an important and missing piece in the human lifespan developmental and aging neuroscience literature, and the authors should be commended for undertaking such an ambitious effort. The novelty and strength of this study is in their focus on studying "the existence of sub-networks that present significant age-related alterations in connectivity weight." There are, however, a number of serious shortcomings and concerns that substantially temper my overall enthusiasm for this report.

First, this appears to be the first work to evaluate longitudinal changes in white matter network properties in older age, which is a major strength. However, the reported sample covers a 30 year age range, but age is not accounted for in the longitudinal analysis. This is critical, because to do otherwise assumes homogeneity of white matter change across the sample, which is not supported by prior longitudinal findings. Moreover, the smaller sample size (N=51) appears insufficient for necessary individual differences analysis of age, as well as sex, but this is difficult to determine without a proper power analysis. I appreciate the description of the apparent limitations of the sample, but it does not mitigate the inherent challenges this poses. Similarly, despite its commendable inclusion of a population-based sample, its specific value in characterizing longitudinal network changes is lost in the absence of at least controlling for established differences in age and biological sex. Thus, without accounting for these necessary confounds, or without comparison data from younger adults, it is difficult to know what conclusions to draw here about aging, or even about age-related changes.

In addition, the exclusively data-driven, exploratory nature of this work poses clear challenges for understanding the potential replicability or endurance of their findings. Both the cross-sectional network analyses they cite, as well as other recent longitudinal studies of diffusion MRI in middle and older age not cited, should provide ample priors for generating testable hypotheses. For example, these findings are consistent with recent longitudinal reports of diffusion tensor data in aging and lifespan samples showing that earlier developing white matter fibers such as those in the inter-hemispheric commissural fibers, change differently from the later-developing intra-hemispheric association fibers. This last-in-first-out framework at least offers some theoretical framework to anchor their changes without relying on either functional or volumetric effects that are not included here. In the introduction the authors cite cross-sectional findings describing age-rated changes, which is logically insufficient support for their hypotheses. Similarly disconnection hypotheses from Geschwind should be cited more clearly, rather than just citing the subsequent studies that utilized this perspective.

For the participants, the cutoff for MMSE exclusion, while following the established norms, do seem rather low. Some discussion of this comparison - have there been any normative data published since the original version in 1994? In addition, it is unclear whether this was only used at baseline, or if the same criteria were applied for excluding participants at follow up.

One primary question concerns how MRI data processing may affect the reliability of the network identification methods they utilized, particularly regarding changes over time in connector hubs. It would be useful to know how strongly this may be affected by other, potentially spurious influences including scanner drift over time since there was no phantom used, or the lack of coregistration between longitudinal occasions. Similarly, 30 direction diffusion MRI data on a 12 channel coil is not optimal for tractography. I also question whether the inverted affine transformation of AAL labels from MNI space reliably captures the nonlinear nature of native-to-template registration with necessary anatomical precision for making longitudinal comparisons regarding the number of streamlines between such regions. The use of the AAL atlas also needs to be more clearly and more strongly justified as this parcellation may not reflect either the most significant volumetric declines or white matter changes, and including this without also evaluating functional connectivity seems confusing.

The description of the statistical analyses would benefit from more detail, particularly the description of the NBS method. I wonder if the authors can expand on how their chosen method for identifying subnetworks is protected against various sources of bias. This is a major issue for longitudinal analyses as issues with measurement reliability can poses key challenges for applying this method independently to the two occasions and to the reported results as reflecting a valid estimate of change. Paired sample t-tests may also limit the study's ability to properly estimate change and these are akin to difference tests on the manifest level, in which measurement error is amplified.

Regarding the results of network modeling, I was unclear if the longitudinal changes in modularity structure, shifting from leftward to rightward lateralization over time was differently influenced by age. The

dramatic nature of this shift was rather alarming, and would benefit considerably from further validation. It would be helpful to know how strongly this might be influenced by other aspects of the study's limitations - insufficient sample size, 1.5T diffusion parameters, etc - could these affect the differences in modularity over time?

I found the discussion somewhat disconnected from the larger question of white matter aging. The authors could better address why regions like the caudate nucleus would lose its connector hub status in aging or how other aspects of the network modeling might affect this, independent of major changes in connectivity? The discussion would benefit from greater integration with other findings of longitudinal changes in white matter in middle and older age, as well as in structural and functional networks.

Other/minor:

As a grammatical note to the authors, I found the narrative well written and easy to follow in most of the manuscript, but there were some systematic issues that should be corrected in a revision. For example, I know many non-native English speakers struggle with the use of definite and indefinite articles 'the,' and 'a,' and the manuscript would benefit from additional revision to ensure expected compliance with basic usage and grammatical rules. Similarly, checking for the appropriateness of prepositions of/from/for would also be helpful for consistency with common English usage.

The results report, "Characteristic path length had significant differences between the two timepoints..." this should be clearly interpreted to understand which occasion was greater, and generally what this represents.

The description of the network construction states, "After this, the upper and lower triangles were averaged, which originated an undirected connectivity matrix." This is not clear and would benefit from rewording or revision.

Authors' Response

Comments to the Author

The current study examines structural connectome metrics (generated from diffusion weighted imaging) in a longitudinal sample of older adults and reports changes in intra-hemispheric connectivity as well as increased segregation over time. The longitudinal nature of this work makes it an important addition to the literature which largely has examined age differences cross-sectionally. My questions and comments largely regard clarification of some of the methods and interpretation of these findings as well as a suggestion to control for potential additional confounds (ie head motion)

1. Methods: Was head-motion during the diffusion scan accounted for in any of the analyses? Prior work has shown (particularly in aging populations that tend to move more in scanner) that increased head motion can selectively different connectome estimates (eg short vs long connection), even after correcting for motion during preprocessing (see Baum et al., 2018, Neuroimage). Do the results hold when statically controlling for motion (ie including frame displacement as a nuisance covariate) in the analyses?

Author: *We are thankful to the reviewer for highlighting this question.*

We have performed head-motion analysis that is now included in the supplementary file (Supplementary Figures 1 and 2). The results of this analysis show that head motion is not significantly correlated with age and also head-motion displacement values did not significantly differ between timepoints. Thus, we concluded that it was not necessary to control for head-motion in the statistical analyses, the motion correction performed in the preprocessing was

sufficient. We have highlighted this in the methods description (please, see page 9, section "MRI Data preprocessing").

2. Methods: For the PCA of the neurocognitive measures, why did the authors include MMSE as a measure (as it is generally considered a more clinical/screening assessment)? Was the total MMSE score used (summed across all questions)? Because the MMSE includes sections that specifically test memory vs more executive functions, it is not clear how total MMSE score would fit into either component of the PCA, so if the authors believe MMSE should be included, it would make more sense to include sub-section scores rather than the total score.

Author: The use of the MMSE was motivated by the fact that it is widely

used for the assessment of cognition in older samples, and by its rapid administration time of the instrument. Furthermore, the MMSE presents good reliability allowing between-studies' comparability. Whereas the MMSE assesses distinct cognitive domains, it is frequently regarded as uni-factorial. In fact, as the reviewer correctly highlighted, the MMSE is generally used as a screening assessment. For that purpose, a global score is typically obtained indicating whether the individual has cognitive impairment.

With this in mind, we replicated previous analytical pipelines of principal component analysis where the total scores of each test were used instead of individual dimensions. Such approach has yielded high reliable main components as previously reported by our group both cross-sectionally (Castanho et al., 2014) and longitudinally (Moreira et al., 2018). Nevertheless, we acknowledge that the nomenclature used for the measure comprising the MMSE (reported as the dimension of executive functioning) may not be the most appropriate. Instead, this component should be described as representing general cognitive status and executive functioning.

3. Methods: The PCA of the neurocognitive measures is not clear - were two separate PCAs conducted per time-point or were data pooled together? The authors also describe z-scored cognitive measures, but it is also not clear how they were generated (z-scoring of the PCA factor scores?). They report mean z-scores that are not equal to 0 (as would be expected from the z-scoring process) which also has me confused about how this was conducted.

Author: We appreciate the reviewer's comment. In fact, the individual scores were obtained from the factor scores estimated from the longitudinal measurement invariance (reported in (Moreira et al., 2018)). We have now described the analytical strategy in the body of the manuscript (please, see page 7, section "Neurocognitive Assessment"). Briefly, the scores on each neurocognitive measure were included in a model where four latent variables were defined: EXEC M1, MEM M1, EXEC M2 and MEM M2. Having provided support for partial strong invariance, the equivalence of the factorial structure and factor loadings for the neurocognitive tests can be assumed. As such, individual factor scores can be obtained for each latent score that will allow the comparison between dimensions between the baseline and the follow-up assessment. Nevertheless, to enable such comparison, the values are not z-score transformed (as the reviewer correctly

identified) and thus should be referred as simply factor mean scores. We have now updated this description in the manuscript.

4. Discussion: The authors largely describe their structural connectome findings in relation to functional connectivity literature, but it would be helpful to also relate their findings to age-effects white matter structure. Do the connections they find map onto major white matter bundles? I find it curious that the authors do not use the term "white matter" at all in the discussion, so are these structural connectivity metrics not meant to represent properties of underlying white matter structure?

Author: We appreciate this relevant comment, which helped to substantially improve the quality of our manuscript. In order to fully address it, we performed an additional analysis where we identified the white matter tracts connecting each pair of regions. The complete description of the analysis and the obtained results is in the methods and results sections of the manuscript (please, see pages 15 and 18, sections "White matter tracts analysis"). In summary, we found that connections with a decrease in structural connectivity were mainly constituted by association fibers (anterior thalamic radiation, cingulum bundle, uncinate fasciculus and superior longitudinal fasciculus), while connections with increasing structural connectivity involved all types of fibers: association, commissural and projection. Since association fibers have a later maturation in comparison to commissural and projection fibers, and we found higher disruption in connectivity in association fibers, we concluded that this finding gives support to the last-in-first-out hypothesis. We also incorporated the interpretation of this finding in the discussion (please, see page 22) and included the term white matter as suggested by the reviewer.

5. Discussion: Similar to the above point, what are the neurobiological changes that might account for the reported reorganization of structural connectivity (ie the finding of increased connectivity estimates)? Are new white matter connections being made? Is myelination along existing connections increasing? Again, the authors never explicitly mention "white matter" in the discussion so it is not entirely clear to me what "structural connectivity" is truly representing at the neurological level.

Author: We acknowledge the reviewer raises an important consideration. Structural connectivity matrices were constructed from diffusion MRI which allows us to measure the apparent diffusion coefficient of water in tissue, thus it is not possible to directly infer white matter physical properties, such as, axon density, caliber and myelination (Jones, 2010; Jbabdi et al., 2015). Probabilistic tractography with dMRI data allows the estimation of a connectivity probability between regions (i.e. the probability that a connection exists between two regions) (Jbabdi and Johansen-Berg, 2011), which is what each element in our structural connectivity matrices represents. The finding of increased connectivity reflects higher probability that an anatomical connection exists between the regions of the sub-network identified (thalamus, putamen, middle/anterior cingulate cortex, supplementary

motor area, precentral gyrus). Although some confounding factors can influence these probabilities, some true anatomical factors can also have a contribution and one example is axon density (Jbabdi et al., 2015). Thus, we can infer that our finding of increased connectivity could be related to an increase in the number of axons connecting the regions of this sub-network. As we stated in the previous reply, we included the term white matter in the discussion as suggested by the reviewer.

6. Discussion: The authors describe the finding of increased connectivity being associated with worse cognition as compensatory. However, recent guidelines put out by many leaders in the field of neurocognitive aging stress that the term compensation should be used to refer to cognition-enhancing brain differences (Cabeza et al., 2019, Nat Rev Neurosci). The authors should reassess how their findings fit in with the terminology put forth in the recent guidelines.

Author: This is an important question. Recent guidelines suggest that two criteria should be fulfilled in order to use the term compensation,

which are a clear evidence of what is being compensated for, that could be either a reduction in neural resources or an increase in task demands or both, and evidence that the enhanced activation is associated with a beneficial effect on cognitive performance. In our study, we did not analyze task-related functional imaging data, thus we cannot directly infer the relation between increased activation and cognitive performance. Our interpretation of the findings was that the decrease in cognitive performance induced a reorganization of brain structural connectivity (i.e. an increase in inter-hemispheric connections) and this reorganization is probably critical for the older adults still be able to perform cognitive tasks, although not in the same levels as before. This is why we termed it compensation, but we acknowledge that it is not a clear direct association and it will need further research to clarify it. We have clarified this in the discussion (please, see page 23).

Reviewer #2

Comments to the Author

The manuscript reports efforts to characterize longitudinal changes in graph theory-based structural brain network properties derived from diffusion MRI data in middle- and older aged adults. This effort represents an important and missing piece in the human lifespan developmental and aging neuroscience literature, and the authors should be commended for undertaking such an ambitious effort. The novelty and strength of this study is in their focus on studying "the existence of sub-networks that present significant age-related alterations in connectivity weight." There are, however, a number of serious shortcomings and concerns that substantially temper my overall enthusiasm for this report.

First, this appears to be the first work to evaluate longitudinal changes in white matter network properties in older age, which is a major strength. However, the reported sample covers a 30 year age range, but age is not accounted for in the longitudinal analysis. This

is critical, because to do otherwise assumes homogeneity of white matter change across the sample, which is not supported by prior longitudinal findings. Moreover, the smaller sample size (N=51) appears insufficient for necessary individual differences analysis of age, as well as sex, but this is difficult to determine without a proper power analysis. I appreciate the description of the apparent limitations of the sample, but it does not mitigate the inherent challenges this poses. Similarly, despite its commendable inclusion of a population-based sample, its specific value in characterizing longitudinal network changes is lost in the absence of at least controlling for established differences in age and biological sex. Thus, without accounting for these necessary confounds, or without comparison data from younger adults, it is difficult to know what conclusions to draw here about aging, or even about age-related changes.

Author: *Thank you for this important comment. In order to fully address*

it, we have re-analyzed the data, which replicated and strengthened our results. We recognize that rates of white matter change are not homogeneous across age and thus can potentially impact the reliability of results. In order to clarify if the identified longitudinal network changes in structural connectivity (SC) were driven by differences in age or sex, we analyzed the consistency of signatures of SC in the different timepoints, i.e., how consistent are the patterns of estimated SC across all subjects in a timepoint, as well as between timepoints. To do this, we used the following two strategies for evaluating timepoint consistency (TC) in SC:

TC-I: Intra-timepoint consistency measured as the Pearson's correlation between each subject's SC and timepoint mean SC (considering upper diagonal matrix elements). The resulting r values were z -transformed (Fisher-Z transformation) before averaging and converting (inverse of Fisher-Z) the resultant timepoint consistency back to r scale. This value represents the within-timepoint consistency, i.e. for each timepoint, how well all subjects' SC correlate with the timepoint's average SC.

TC-II: Intra-timepoint consistency measured as the distribution of Pearson's correlations between all possible pairs of subjects in a timepoint. The resulting distribution of all pairwise (pairs of subjects) SC comparisons is represented as a histogram. This indicates how well SCs in a timepoint correlate with each other. Inter-timepoint consistency was also assessed by considering all subjects as part of the same timepoint.

Concerning TC-I, i.e. intra-timepoint consistency measured as the association between individual SC signatures and timepoint average SC, we confirmed that both timepoints reveal a very high level of intratimepoint consistency (M1: 97.6%; M2: 97.5%).

Regarding TC-II, i.e. degree of association between the signatures of SC for all pairs of subjects in the same timepoint, we show again a high level of timepoint consistency in SC, for both timepoints (100% and 99.8% of all pairwise combinations in M1 and M2 timepoints

respectively have a correlation higher than $r=0.9137$, with number of occurrences peaking at approximately $r=0.96$ for both timepoints). The overlap between the distributions of intra-timepoint consistency of both timepoints is additionally confirmed by the inter-timepoint consistency distribution (M1-M2: peak at approximately $r=0.95$). This suggests that, at a global level, the patterns of SC are highly consistent within and between timepoints, and thus potential differences due to age and sex do not have a significant impact on the estimation of SC patterns.

In summary, this supplementary analysis suggests that longitudinal network differences are not driven by differences in age and sex, which strengthens the reliability of the reported findings. Additionally, the high level of inter-timepoint consistency could explain our lack of significant longitudinal differences in global topological metrics. It gives support to the idea that relevant topological differences are limited to sub-networks or individual nodes.

We added the above information concerning timepoint consistency in the methods (page 10), results (page 16) and supplementary material (supplementary figure 7) sections.

In addition, the exclusively data-driven, exploratory nature of this work poses clear challenges for understanding the potential replicability or endurance of their findings. Both the cross-sectional network analyses they cite, as well as other recent longitudinal studies of diffusion MRI in middle and older age not cited, should provide ample priors for generating testable hypotheses. For example, these findings are consistent with recent longitudinal reports of diffusion tensor data in aging and lifespan samples showing that earlier developing white matter fibers such as those in the interhemispheric commissural fibers, change differently from the laterdeveloping intra-hemispheric association fibers. This last-in-firstout framework at least offers some theoretical framework to anchor their changes without relying on either functional or volumetric effects that are not included here. In the introduction the authors cite cross-sectional findings describing age-rated changes, which is logically insufficient support for their hypotheses. Similarly disconnection hypotheses from Geschwind should be cited more clearly, rather than just citing the subsequent studies that utilized this perspective.

Author: Following the reviewer's suggestion, we included in the introduction references for longitudinal studies of diffusion MRI that could provide priors for our hypothesis, but in some cases (e.g. topological metrics) only cross-sectional studies exist so far (please, see page 5, section "Introduction"). Further, we also incorporated the last-in-first-out framework as a prior for the hypothesis of differential age-related changes in distinct white matter tracts and also included it in the discussion (please, see page 5, section "Introduction" and page 21, section "Discussion"). We have also included the citation of the original paper on the disconnection hypothesis from Geschwind (please, see page 3, section

"Introduction").

For the participants, the cutoff for MMSE exclusion, while following the established norms, do seem rather low. Some discussion of this comparison - have there been any normative data published since the original version in 1994? In addition, it is unclear whether this was only used at baseline, or if the same criteria were applied for excluding participants at follow up.

Author: *Previous studies have reported that age and education explain a considerable part of the variation of the MMSE (12% according to (Bravo and Hebert, 1997)). The lower levels of education of our sample (which are representative of the Portuguese older population) thus may explain the low scores on this test. The same criteria were applied for excluding participants at follow-up and we have included this statement in the methods description (please, see page 6, section "Participants").*

One primary question concerns how MRI data processing may affect the reliability of the network identification methods they utilized, particularly regarding changes over time in connector hubs. It would be useful to know how strongly this may be affected by other, potentially spurious influences including scanner drift over time since there was no phantom used, or the lack of coregistration between longitudinal occasions. Similarly, 30 direction diffusion MRI data on a 12 channel coil is not optimal for tractography. I also question whether the inverted affine transformation of AAL labels from MNI space reliably captures the nonlinear nature of native-to-template registration with necessary anatomical precision for making longitudinal comparisons regarding the number of streamlines between such regions. The use of the AAL atlas also needs to be more clearly and more strongly justified as this parcellation may not reflect either

the most significant volumetric declines or white matter changes, and including this without also evaluating functional connectivity seems confusing.

Author: *We acknowledge the reviewer for raising these relevant points. It is our understanding that MRI data processing and other artifacts, such as scanner drift, had minimal influence in the pattern of our findings, since MRI data processing, data acquisition protocols and MRI scanner were exactly the same for both timepoints. Previous work have examined the reproducibility of DTI measurements, structural connectome and graph theory metrics and all of these had high reproducibility with data from the same scanner (Takao et al., 2011; Bonilha et al., 2015). Coregistration between longitudinal assessments was not considered since probabilistic tractography was performed in the subject space instead of a common space. Regarding the 30 direction*

diffusion MRI data and the 12-channel head coil, we acknowledge that it is not optimal for tractography purposes, but it was the only equipment available for this study. Furthermore, we used the affine transformation of AAL labels from MNI space to the subjects' diffusion space, since this registration yielded better results in comparison to the nonlinear registration (see figures 1 and 2 below). Concerning

the choice of the AAL atlas, since in this study functional data was also collected, we opted for this atlas to allow future comparisons between structural and functional data. Furthermore, this atlas has been previously used in studies focusing only in structural brain networks (Iturria-Medina et al., 2008; Gong et al., 2009).

Figure 1 – Example of AAL template affine registration to a subject's diffusion space

Figure 2 – Example of AAL template nonlinear registration to a subject's diffusion space

The description of the statistical analyses would benefit from more detail, particularly the description of the NBS method. I wonder if the authors can expand on how their chosen method for identifying subnetworks is protected against various sources of bias. This is a major issue for longitudinal analyses as issues with measurement reliability can pose key challenges for applying this method independently to the two occasions and to the reported results as reflecting a valid estimate of change. Paired sample t-tests may also limit the study's ability to properly estimate change and these are akin to difference tests on the manifest level, in which measurement error is amplified.

Author: We appreciate the reviewer's comment. The NBS method was not applied independently to the two occasions. Instead we specified a model in terms of the General Linear Model (GLM) to perform a paired sample t-test. This was done by giving as inputs to the NBS a design matrix and a contrast vector. This is now clarified in the methods section (please, see page 13, section "Statistical Analysis").

Additionally, as it was described in the methods the NBS method allows the identification of significantly different sub-networks, while controlling for the family-wise error rate (FWER). Since our structural connectivity networks have 90 nodes, the total number of possible edges is $90 \times 89 / 2 = 40005$. Testing the hypothesis of interest at the edge level, thus poses a multiple comparisons problem. NBS allows to deal with this problem, by first testing the hypothesis at each edge and finding sub-networks constituted by interconnected edges that survived a user defined primary threshold. Then, the significance of these sub-networks is calculated through permutation testing.

Regarding the results of network modeling, I was unclear if the longitudinal changes in modularity structure, shifting from leftward to rightward lateralization over time was differently influenced by age. The dramatic nature of this shift was rather alarming, and would benefit considerably from further validation. It would be helpful to know how strongly this might be influenced by other aspects of the study's limitations - insufficient sample size, 1.5T diffusion parameters, etc - could these affect the differences in modularity over time?

Author: We thank the reviewer for raising important questions. Our analysis were performed on longitudinal data (i.e. the same individuals were evaluated at two timepoints) and computed the modularity structure for both timepoints. Thus, the observed change in modularity is due to the effect of time (i.e. how the modularity organization evolved along a mean time of 54 months) and we did not explore the effect of age (i.e. if the modularity changes were different for distinct ages) as it was not our main interest. Also,

the limitations of our study (1.5T scanner, low sample size and period of evaluation) could affect the obtained results in modularity and we have reinforced this in the discussion (please, see page 28).

I found the discussion somewhat disconnected from the larger question of white matter aging. The authors could better address why regions like the caudate nucleus would lose its connector hub status in aging or how other aspects of the network modeling might affect this, independent of major changes in connectivity? The discussion would benefit from greater integration with other findings of longitudinal changes in white matter in middle and older age, as well as in structural and functional networks.

Author: *A connector hub is defined based on the node's connectivity distribution within and between modules. Particularly, a connector hub is a node with high within-module degree z-score and high participation*

coefficient, meaning that it will have many connections with other modules and thus plays a key role in inter-modular communication. Left caudate and right midcingulate cortex are part of the sub-network that had decreasing connectivity along time. Hence, this decrease in connectivity may have contributed to the loss of its role as a connector hub. We have extended the discussion to include this (please,

see page 25). We have included an additional analysis of white matter fibers involved in the connections of each sub-network following one of the comments of reviewer 1. Now, in the discussion we included findings of longitudinal changes in white matter (please, see page 22).

Other/minor:

As a grammatical note to the authors, I found the narrative well written and easy to follow in most of the manuscript, but there were some systematic issues that should be corrected in a revision. For example, I know many non-native English speakers struggle with the use of definite and indefinite articles 'the,' and 'a,' and the manuscript would benefit from additional revision to ensure expected compliance with basic usage and grammatical rules. Similarly, checking for the appropriateness of prepositions of/from/for would also be helpful for consistency with common English usage.

Author: *A native speaker has revised the manuscript in order to correct*

these issues. We have implemented several modifications and we consider that the text is now clearer.

The results report, "Characteristic path length had significant differences between the two timepoints..." this should be clearly interpreted to understand which occasion was greater, and generally what this represents.

Author: *We have changed the description of this result (please, see page 18, section "Topological Organization Longitudinal Changes") and added additional information in the discussion (please, see page 23) that interprets the result found.*

The description of the network construction states, "After this, the upper and lower triangles were averaged, which originated an

undirected connectivity matrix." This is not clear and would benefit from rewording or revision.

Author: *We have modified this description to make it clearer (please, see page 9, section "Network Construction").*

References

- Bonilha, L., Gleichgerrcht, E., Fridriksson, J., Rorden, C., Breedlove, J. L., Nesland, T., et al. (2015). Reproducibility of the Structural Brain Connectome Derived from Diffusion Tensor Imaging. *PLOS ONE* 10, e0135247. doi:10.1371/journal.pone.0135247.
- Bravo, G., and Hebert, R. (1997). Age- and education-specific reference values for the Mini-Mental and Modified Mini-Mental State Examinations derived from a non-demented elderly population. *International Journal of Geriatric Psychiatry* 12, 1008-1018. doi:10.1002/(SICI)1099-1166(199710)12:10<1008::AIDGPS676>3.0.CO;2-A.
- Castanho, T. C., Moreira, P. S., Portugal-Nunes, C., Novais, A., Costa, P. S., Palha, J. A., et al. (2014). The role of sex and sex-related hormones in cognition, mood and well-being in older men and women. *Biological Psychology* 103, 158-166. doi:10.1016/j.biopsycho.2014.08.015.
- Gong, G., Rosa-Neto, P., Carbonell, F., Chen, Z. J., He, Y., and Evans, A. C. (2009). Age- and gender-related differences in the cortical anatomical network. *The Journal of Neuroscience* 29, 15684-15693. doi:10.1523/JNEUROSCI.2308-09.2009.
- Iturria-Medina, Y., Sotero, R. C., Canales-Rodríguez, E. J., Alemán-Gómez, Y., and Melie-García, L. (2008). Studying the human brain anatomical network via diffusion-weighted MRI and Graph Theory. *NeuroImage* 40, 1064-1076. doi:10.1016/j.neuroimage.2007.10.060.
- Jbabdi, S., and Johansen-Berg, H. (2011). Tractography: Where Do We Go from Here? *Brain Connectivity* 1, 169-183. doi:10.1089/brain.2011.0033.
- Jbabdi, S., Sotiropoulos, S. N., Haber, S. N., Van Essen, D. C., and Behrens, T. E. (2015). Measuring macroscopic brain connections in vivo. *Nature Neuroscience* 18, 1546-1555. doi:10.1038/nn.4134.
- Jones, D. K. (2010). Challenges and limitations of quantifying brain connectivity in vivo with diffusion MRI. *Imaging in Medicine* 2, 341-355. doi:10.2217/iim.10.21.
- Moreira, P. S., Santos, N., Castanho, T., Amorim, L., Portugal-Nunes, C., Sousa, N., et al. (2018). Longitudinal measurement invariance of memory performance and executive functioning in healthy aging. *PLOS ONE* 13, e0204012. doi:10.1371/journal.pone.0204012.
- Takao, H., Hayashi, N., and Ohtomo, K. (2011). Effect of scanner in asymmetry studies using diffusion tensor imaging. *NeuroImage* 54, 1053-1062. doi:10.1016/j.neuroimage.2010.09.023.

2nd Editorial Decision

Dear Dr Coelho:

Thank you for submitting your manuscript to the Journal of Neuroscience Research. We have now received the reviewer feedback and have appended those reviews below. I am glad to say that the reviewers are overall very enthusiastic and supportive of the study. They did raise some concerns and made some suggestions for clarification, but I expect that these points should be relatively straightforward to address. If there are any questions or points that are problematic, please feel free to contact me. I will be glad to discuss.

We ask that you return your manuscript within 30 days. Please explain in your cover letter how you have changed the present version and submit a point by point response to the editors' and reviewers' comments. If you require longer than 30 days to make the revisions, please contact Dr Junie Warrington (jpwarrington@umc.edu). To submit your revised manuscript: Log in by clicking on the link below

(If the above link space is blank, it is because you submitted your original manuscript through our old submission site. Therefore, to return your revision, please go to our new submission site here (submission.wiley.com/jnr) and submit your revision as a new manuscript; answer yes to the question "Are you returning a revision for a manuscript originally submitted to our former submission site (ScholarOne Manuscripts)? If you indicate yes, please enter your original manuscript's Manuscript ID number in the space below" and including your original submission's Manuscript ID number (jnr-2020-Jul-8925.R1) where indicated. This will help us to link your revision to your original submission.)

Thank you again for your submission to the Journal of Neuroscience Research; we look forward to reading your revised manuscript.

Best Wishes,

Dr Jeremy Hogeveen
Associate Editor, Journal of Neuroscience Research

Dr Junie Warrington
Editor-in-Chief, Journal of Neuroscience Research

Editorial Comments to Author:

1. Please increase the font size in each of the figures. As presented, the text is very hard to read without zooming in significantly. We also ask that a single font style is used throughout the manuscript.
2. Please upload a graphical abstract, which we are asking of all authors submitting original research articles. This is intended to provide readers with a visual representation of the conclusions and an additional way to access the contents and appreciate the main message of the work. What we require is a .tif image file and a .doc text file containing an abbreviated abstract. For the image, labels, although useful, must be kept to a minimum and the image should be 400 x 300, 300 x 400, or 400 x 400 pixels square and at a resolution of 72 dpi. This can be one of the figures from your article, or something slightly different, as long as it represents your study. Instructions for this can be found in our author guidelines online at [http://onlinelibrary.wiley.com/journal/10.1002/\(ISSN\)1097-4547/homepage/ForAuthors.html](http://onlinelibrary.wiley.com/journal/10.1002/(ISSN)1097-4547/homepage/ForAuthors.html)
3. Please move all methods included in the supplementary materials to the main manuscript. JNR does not have a limit on the length of manuscripts or the number of figures included in manuscripts. If supplementary materials are maintained, please include an additional title page since the supplementary materials are stored separately from the main manuscript.

Associate Editor: Hogeveen, Jeremy

Comments to the Author:

Dear Dr. Sousa,

Thank you for revising and resubmitting your work to Journal of Neuroscience Research. Myself and one of the original expert reviewers reviewed the revised submission, and we both agree that you have done a tremendous job addressing many of the original issues with the submission. However, we agree that there are still several substantive issues with the submission regarding the interpretation of your results. If you feel that you can address these remaining issues, we would welcome a revised version of the manuscript.

Best Wishes,

Dr Jeremy Hogeveen

Associate Editor, Journal of Neuroscience Research

Reviewer: 2

Comments to the Author

In their revised manuscript, Coelho and colleagues have commendably addressed the majority of comments from the two anonymous peer reviewers. However, there are several smaller issues that should also be addressed:

The abstract would benefit from a more tempered tone - e.g., "disruption of this communication is thought to account for the age-related deterioration observed in cognition," should be clarified as only one potential contributor to age-related cognitive declines; it would help to have clarification that 'structural connectivity' specifically refers to white matter fiber connections, as others have looked at covariance of gray matter regional volume or thickness to also derive a different representation of the structural connectome.

I would caution against the use of interpreting tensor-parameter findings as reflecting 'white matter integrity.'

I commend the authors for the inclusion of additional information on how the cognitive factors were generated. However, it would also be helpful to have further clarification on the method for outputting the factor scores and the software used for the CFA.

Anatomically, I'm unclear regarding the structural connection between the left caudate nucleus and the right cingulate gyrus. Is this connection mediated via colossal fibers? Further discussion would be helpful to understand the anatomical specificity of these changes.

The authors report that characteristic path length was the only significant network metric that changed over time. However, the manuscript would benefit from more direct interpretation of this finding. The suggestion that no other changes in network properties may be taking place seems confounded by the limited sample size and coverage of the adult lifespan. Indeed, other studies to which they compare their findings have larger samples, which at the very least needs to be addressed as a limitation. Given the reasonably long duration between measurements, one concern is the lack of clear individual differences in change could simply reflect an underpowered sample, rather than as confirmation of invariance in change. In addition, the focus of the interpretation in light of compensation needs to be supported more definitively with findings

from the repeated behavioral measurements. That is, the basis of evidence for this relationships should be to show differential change in network properties in those whose cognitive scores decline vs. remain stable.

I have some misgivings regarding the choice to interpret the nonsignificant trend in the relationship between cognitive scores and mean sub-network SC values. The purpose of correcting results of null hypothesis significance testing for multiple comparisons is to mitigate spurious results. If the authors are convinced that this is simply a result of an overly-conservative adjustment, then additional evidence to support this would be useful (i.e., showing bootstrap resampling [i.e., 1000 draws or more] of 95% confidence intervals results in a significant effect); otherwise, this interpretation seems unwarranted in the absence of a robust and significant effect following corrections.

Authors' Response

Associate Editor: Hogeveen, Jeremy Comments to the Author: Dear Dr. Sousa, Thank you for revising and resubmitting your work to Journal of Neuroscience Research. Myself and one of the original expert reviewers reviewed the revised submission, and we both agree that you have done a tremendous job addressing many of the original issues with the submission. However, we agree that there are still several substantive issues with the submission regarding the interpretation of your results. If you feel that you can address these remaining issues, we would welcome a revised version of the manuscript. Best Wishes, Dr Jeremy Hogeveen Associate Editor, Journal of Neuroscience Research

Reviewer: 2 Comments to the Author In their revised manuscript, Coelho and colleagues have commendably addressed the majority of comments from the two anonymous peer reviewers. However, there are several smaller issues that should also be addressed: The abstract would benefit from a more tempered tone - e.g., "disruption of this communication is thought to account for the age-related deterioration observed in cognition," should be clarified as only one potential contributor to age-related cognitive declines; it would help to have clarification that 'structural connectivity' specifically refers to white matter fiber connections, as others have looked at covariance of gray matter regional volume or thickness to also derive a different representation of the structural connectome. I would caution against the use of interpreting tensor-parameter findings as reflecting 'white matter integrity.' I commend the authors for the inclusion of additional information on how the cognitive factors were generated. However, it would also be helpful to have further clarification on the method for outputting the factor scores and the software used for the CFA. Anatomically, I'm unclear regarding the structural connection between the left caudate nucleus and the right cingulate gyrus. Is this connection mediated via colossal fibers? Further discussion would be helpful to understand the anatomical specificity of these changes. The authors report that characteristic path length was the only significant network metric that changed over time. However, the manuscript would benefit from more direct interpretation of this finding. The suggestion that no other changes in network properties may be taking place seems confounded by the limited sample size and coverage of the adult lifespan. Indeed, other studies to which they compare their findings have larger samples, which at the very least needs to be addressed as a limitation. Given the reasonably long duration between measurements, one concern is the lack of clear individual differences in change could simply reflect an underpowered sample, rather than as confirmation of invariance in change. In addition, the focus of the interpretation in light of compensation needs to be supported more definitively with findings from the repeated behavioral measurements. That is, the basis of evidence for this relationships should be to show differential change in network properties in those whose cognitive scores decline vs. remain stable. I have some misgivings regarding the choice to interpret the nonsignificant trend in the relationship between cognitive scores and mean sub-network SC values. The purpose of correcting results of null hypothesis significance testing for multiple comparisons is to mitigate spurious results. If the authors are convinced that this is simply a result of an overly-conservative adjustment, then additional evidence to support this would be useful (i.e., showing bootstrap resampling [i.e., 1000 draws or more] of 95% confidence intervals results in a significant effect); otherwise, this interpretation seems unwarranted in the absence of a robust and significant effect following corrections.

3rd Editorial Decision

Dear Dr Coelho:

Thank you for submitting your manuscript "Reorganization of brain structural networks in aging: a

longitudinal study" by Coelho, Ana; Fernandes, Henrique; Magalhães, Ricardo; Moreira, Pedro; Marques, Paulo; Soares, José; Amorim, Liliana; Portugal-Nunes, Carlos; Castanho, Teresa; Santos, Nadine; Sousa, Nuno.

You will be pleased to know that your manuscript has been accepted for publication. Thank you for submitting this excellent work to our journal.

In the coming weeks, the Production Department will contact you regarding a copyright transfer agreement and they will then send an electronic proof file of your article to you for your review and approval.

Please note that your article cannot be published until the publisher has received the appropriate signed license agreement. Within the next few days, the corresponding author will receive an email from Wiley's Author Services asking them to log in. There, they will be presented with the appropriate license for completion. Additional information can be found at <https://authorservices.wiley.com/author-resources/Journal-Authors/licensing-open-access/index.html>

Would you be interested in publishing your proven experimental method as a detailed step-by-step protocol? Current Protocols in Neuroscience welcomes proposals from prospective authors to disseminate their experimental methodology in the rapidly evolving field of neuroscience. Please submit your proposal here: <https://currentprotocols.onlinelibrary.wiley.com/hub/submitproposal>

Congratulations on your results, and thank you for choosing the Journal of Neuroscience Research for publishing your work. I hope you will consider us for the publication of your future manuscripts.

Sincerely,

Dr Jeremy Hogeveen
Associate Editor, Journal of Neuroscience Research

Dr Junie Warrington
Editor-in-Chief, Journal of Neuroscience Research